# Evaluation of UV-visible MAX-DOAS aerosol profiling products by comparison with ceilometer, sun photometer, and in situ observations in Vienna, Austria

**Stefan F. Schreier[1], Tim Bösch[2], Andreas Richter[2], Kezia Lange[2], Michael Revesz[1], Philipp Weihs[1], Mihalis Vrekoussis[2,3], and Christoph Lotteraner[4]**

[1]Institute of Meteorology and Climatology, University of Natural Resources and Life Sciences, Vienna, Austria

[2]Institute of Environmental Physics, University of Bremen, Germany

[3]Climate and Atmosphere Research Center (CARE-C), The Cyprus Institute, Cyprus

[4]Central Institute for Meteorology and Geodynamics, Vienna, Austria

Correspondence to: S. F. Schreier (stefan.schreier@boku.ac.at)

## Abstract

Since May 2017 and August 2018, two ground-based MAX-DOAS (Multi AXis Differential Optical Absorption Spectroscopy) instruments have been continuously recording daytime spectral UV-visible measurements in the north-west (University of Natural Resources and Life Sciences (BOKU) site) and south (Arsenal site), respectively, of the Vienna city centre (Austria). In this study, vertical aerosol extinction (AE) profiles, aerosol optical depth (AOD), and near-surface AE are retrieved from MAX-DOAS measurements recorded on cloud-free days applying the Bremen Optimal estimation REtrieval for Aerosols and trace gaseS (BOREAS) algorithm. Measurements of atmospheric profiles of pressure and temperature obtained from routinely performed sonde ascents are used to calculate box-air-mass-factors and weighting functions for different seasons. The performance of BOREAS was evaluated against co-located ceilometer, sun photometer, and in situ instrument observations covering all four seasons. The results show that the vertical AE profiles retrieved from the BOKU UV-visible MAX-DOAS observations are in very good agreement with data from the co-located ceilometer, reaching correlation coefficients ($R$) of 0.936-

0.996 (UV) and 0.918-0.999 (visible) during fall, winter, and spring seasons. Moreover, AE extracted using the lowest part of MAX-DOAS vertical profiles (up to 100 m above ground) are highly consistent with near-surface ceilometer AE ($R > 0.865$ and linear regression slopes of 0.815-1.21) during the fall, winter, and spring seasons. A strong correlation is also found for the BOREAS-based AODs when compared to the AERONET ones. Notably, the highest correlation coefficients ($R = 0.953$ and $R = 0.939$ for UV and visible, respectively) were identified for the fall season. While high correlation coefficients are generally found for the fall, winter, and spring seasons, the results are less reliable for measurements taken during summer. For the first time, the spatial variability of AOD and near-surface AE over the urban environment of Vienna is assessed by analyzing the retrieved and evaluated BOREAS aerosol profiling products in terms of different azimuth angles of the two MAX-DOAS instruments and for different seasons. We found that the relative differences of averaged AOD between different azimuth angles are 7-13%, depending on the season. Larger relative differences of up to 32% are found for near-surface AE in the different azimuthal directions. This study revealed the strong capability of BOREAS to retrieve AE profiles, AOD, and near-surface AE over urban environments and demonstrated its use for identifying the spatial variability of aerosols, in addition to the temporal variation.

## 1 Introduction

Atmospheric aerosols are defined as particles (liquid or solid) suspended in the air, with particle diameters in the range of $10^{-9}$ to $10^{-4}$ m (0.001 μm to 100 μm) and various shapes, chemical compositions, and hygroscopic and optical properties (Seinfeld and Pandis, 2006). Aerosols are an important component of the Earth's atmosphere and play a crucial role in atmospheric chemistry, cloud formation and lifetime, Earth's radiation budget, and climate (IPCC, 2013). It has also been widely documented that enhanced atmospheric aerosol loading has adverse effects on human health (Cohen et al., 2005; Liu et al., 2009; Russell and Brunekreef, 2009; Fann et al., 2012; Lelieveld et al., 2015).

Sources of aerosols include natural emissions from the sea surface, soils, terrestrial vegetation, volcanoes, and wildfires, as well as anthropogenic emissions from agricultural and industrial activities, combustion processes, abrasion, and solvent use (Kanakidou et al., 2018).

A number of instruments for ground-based observations have been developed in the last decades to obtain aerosol optical properties and vertical profiles in the troposphere. In situ instruments (e.g. optical particle counters) are often designed to measure particulate matter (PM) concentrations (e.g. PM2.5 and PM10, whose size is defined as less than 2.5 and 10 μm in diameter, respectively) in ambient air.

In addition to in situ measurement techniques, ground-based remote sensing instruments such as sun photometers, LIght Detection And Ranging (LIDAR), ceilometers, and Multi AXis Differential Optical Absorption Spectrometers (MAX-DOAS) as well as corresponding retrieval approaches have been developed to obtain aerosol optical properties and vertical profiles (Holben et al., 1998; Ansmann et al., 2011; Madonna et al., 2018; Frieß et al., 2016).

From sun photometer measurements, precise information on the total aerosol extinction (AE) and scattering phase function can be derived and column-averaged aerosol size distribution, single scattering albedo, and refractive index can be extracted (Holben et al., 1998). A large part of such sun photometer measurement efforts is done in the framework of the AERONET (AErosol RObotic NETwork) network (https://aeronet.gsfc.nasa.gov/).

Research grade LIDARs provide vertical profiles of aerosol backscattering and other information at high vertical resolution (e.g. Madonna et al., 2018). In contrast, the retrieval of attenuated backscatter and aerosol backscattering coefficient from ceilometer observations is limited by instrument accuracy and highly dependent on the availability of data from co-located ancillary instruments (e.g. sun photometer and/or Raman multi-wavelength LIDAR). However, the lower costs and lower maintenance requirements associated with commercial ceilometers make these instruments attractive for ground-based observations of aerosol optical properties and vertical profiles in global scale networks (Madonna et al., 2018; Lotteraner and Piringer, 2016; Baumann-Stanzer et al., 2019).

More than 15 years ago, a method to derive aerosol optical properties and vertical profiles from MAX-DOAS observations was presented (Wagner et al., 2004), and since then has received increasing attention (e.g. Frieß et al., 2006; Clémer et al., 2010; Yilmaz, 2012; Wang et al., 2013; Vlemmix et al., 2015; Chan et al., 2017; Bösch et al., 2018; Beirle et al., 2019; Friedrich et al., 2019). The derived aerosol information has been used for environmental studies as well as for the

validation of satellite observations and model simulations (e.g. Ma et al., 2013). State-of-the-art

MAX-DOAS retrieval algorithms (Tirpitz et al., 2021 and references therein) can be used to

quantify horizontal inhomogeneities in aerosol loading over urban and rural areas, in addition to

the aerosol vertical distribution. Further research efforts are needed to better retrieve aerosol optical

properties and vertical profiles by using the above mentioned algorithms and to compare and

validate the resulting aerosol products against independent co-located measurements.

In this study, we evaluate and analyze AE profiles, aerosol optical depth (AOD), and near-surface

(e.g. the lowest extinction point representing the altitude range between the surface and up to 100

9     m) AE retrieved from UV-visible spectral measurements collected with two MAX-DOAS

instruments in Vienna, Austria, located in the north-west and south of the city center. The retrieval

of UV-visible aerosol profiling products is based on the BOREAS algorithm (Bremen Optimal

estimation REtrieval for Aerosols and trace gaseS) (Bösch et al., 2018), which has been developed

by the Institute of Environmental Physics of University of Bremen (IUP-B) to improve an earlier

profile retrieval algorithm (Wittrock, 2006). The retrieval performance of BOREAS was recently

assessed from synthetic data computed with SCIATRAN (Rozanov et al., 2014) as well as using

real-world measurements taken in September 2016 during the Second Cabauw Intercomparison

campaign for Nitrogen Dioxide measuring Instruments (CINDI-2) in a rural environment (Frieß et

al., 2019; Tirpitz et al., 2021). From spectral measurements collected during CINDI-2, AE profiles

and AOD retrieved with BOREAS were validated with ancillary data. Overall, the results show a

satisfactory performance of BOREAS when the retrieved synthetic profile is close to the a priori.

However, due to the coarse vertical resolution (100 m), a comparison with surface in situ

measurements remains challenging. More recently, Gratsea et al. (2020) reported on the BOREAS

retrieval of AE profiles from ground-based MAX-DOAS measurements taken over the urban

environment of Athens, Greece. For validation purposes, they selected four case studies covering

different seasons and origins of aerosol loads and assessed the performance of BOREAS through

comparison with ground-based lidar AE profiles and sun photometer AOD measurements.

This study aims to evaluate UV-visible aerosol profiling products retrieved with BOREAS, in this

case over the urban environment of Vienna, Austria, through comparison with co-located

instruments. In a first step, AE profiles, AOD, and near-surface AE are retrieved from MAX-DOAS

UV and visible spectral measurements conducted on the roof of a campus building of the University

of Natural Resources and Life Sciences (BOKU) on cloud-free days in the period between September 2017 and August 2019. The BOREAS aerosol profiling products are then compared with data from co-located ceilometer, sun photometer, and in situ instruments. In the second step, additional insights into the spatio-temporal variability of AOD and near-surface AE over the urban environment of Vienna are provided by analyzing BOREAS aerosol profiling products retrieved from measurements collected with two MAX-DOAS instruments (BOKU and Arsenal). By plotting AOD and near-surface AE against simultaneous BOREAS retrievals of tropospheric nitrogen dioxide vertical column densities ($NO_2$ VCDs) and near-surface $NO_2$, respectively, the origin of the aerosol is discussed.

The paper is structured as follows: in section 2, the instruments used in this study and the respective data retrievals/data products are presented. As the study is based on cloud-free days, the methodology to select such days is also introduced in this section. Results and insights into the spatial and temporal patterns and the origin of aerosols over the urban environment of Vienna are presented in section 3, followed by a summary and conclusions (section 4).

## 2 Methodology

### 2.1 Instrumentation

#### 2.1.1 MAX-DOAS

Within the framework of the VINDOBONA (VIenna horizontal aNd vertical Distribution OBservations Of Nitrogen dioxide and Aerosols) project, three ground-based MAX-DOAS instruments have been assembled and put in continuous operation since December 2016, May 2017, and August 2018 at three different locations in Vienna (www.doas-vindobona.at). As the lowest elevation angles which are essential for the retrieval of AE profiles are partially blocked by trees and buildings at the University of Veterinary Medicine (VETMED) site, measurements of the third MAX-DOAS instrument are not considered in this study.

Briefly, MAX-DOAS is a ground-based remote sensing technique for retrieving tropospheric trace gases and aerosols by measuring scattered sunlight at different azimuthal and elevation angles (e.g.

Wagner et al., 2004). The MAX-DOAS systems measuring in Vienna were developed at the Institute of Environmental Physics of the University of Bremen (IUP-B) in Bremen, Germany (e.g. Peters, 2013) and continuously improved during international measurement campaigns such as CINDI, TransBrom, SHIVA, MAD-CAT, and CINDI-2 (Roscoe et al., 2010; Peters et al., 2012; Schreier et al., 2015; Wang et al., 2017; Donner et al., 2019). After the assembly, characterization and testing phases in the laboratory of IUP-B, these instruments were transferred to the locations in Vienna, where they continuously measure scattered sunlight at selected azimuthal and elevation angles to cover air masses over large parts of the urban environment (Schreier et al., 2020).

In this study, UV-visible spectral measurements are taken from the BOKU and Arsenal MAX-DOAS instruments located in the north-west (48.2379°N, 16.3317°E, 267 m a.s.l.) and south (48.1818°N, 16.3908°E, 333 m a.s.l.) of the city center, respectively (see Fig. 1 and Table 1). The two instruments are operating in two configurations: (1) elevation scans at fixed azimuthal directions and (2) azimuthal scans at fixed elevation angles. The former configuration, which is considered in this study, is based on five azimuthal viewing directions between 74° and 144° (BOKU MAX-DOAS) and six azimuthal viewing directions between 324° and 20° (Arsenal MAX-DOAS) (see Fig. 1), which were selected to capture the city center as well as point into the direction of the other MAX-DOAS instruments. Elevation sequences consisting of $\alpha = 0°$, $1°$, $2°$, $3°$, $4°$, $5°$, $10°$, $15°$, $30°$, and $90°$ (zenith) are continuously performed at these azimuthal directions. Measurements taken at $\alpha = 0°$ elevation angle are however not considered for the MAX-DOAS profile retrieval. Further technical details about the spectrometers of the BOKU (Shamrock SR-193i-A) and Arsenal (AvaSpec-ULS2048x64) MAX-DOAS instruments can be found in Schreier et al. (2020) and Behrens et al. (2019), respectively.

### 2.1.2 Ceilometer

The national weather service in Austria, "Zentralanstalt für Meteorologie und Geodynamik" (ZAMG), operates a commercial ceilometer of the type Vaisala CL51 at the site "Hohe Warte" (48.2483°N, 16.3564°E, 198 m a.s.l.) of ZAMG (see Fig. 1 and Table 1), performing routine measurements since July 2012 (Lotteraner and Piringer, 2016). Briefly, the Vaisala CL51 ceilometer (hereinafter referred to as ceilometer) uses diode-laser lidar technology that emits

powerful laser pulses with wavelengths of 910±10 nm in a vertical direction. Backscatter signals are collected from about 50 m above ground up to an altitude of 15 km with a vertical resolution of 10 m (Wagner and Schäfer, 2015). Recently, a method to obtain time series of mixing-heights from ceilometer measurements was developed at ZAMG (Lotteraner and Piringer, 2016). In our study, backscatter profiles with a temporal resolution of about half a minute are converted into AE profiles (see section 2.2.3), which are used to evaluate AE profiles retrieved with BOREAS (see section 3.1.1).

### 2.1.3 Sun photometer

Since May 2016, the Institute of Meteorology and Climatology of BOKU (BOKU-Met) operates a sun photometer (Cimel CE318) within the AERONET project. Briefly, the ground-based Cimel CE318 sun photometer (hereinafter referred to as sun photometer) measures direct sunlight at different selected wavelength ranges. The extinction measurements are used to calculate column-integrated AODs and Angstrom exponents (Holben et al., 1998). Additionally, column-integrated aerosol parameters such as size distribution, refractive index, single scattering albedo, and phase function can be retrieved by applying AERONET Version 3 inversion algorithms. The sun photometer is located on the BOKU-Met measurement platform at a distance of about 2.5 m from the BOKU MAX-DOAS instrument (see Fig. 1 and Table 1). In this study, AOD at 340, 380, 440, 500, 870, and 1020 nm are used for the scaling of ceilometer backscatter profiles (see section 2.2.3) as well as for the comparison with AOD retrieved from BOKU MAX-DOAS UV-visible spectral measurements (see section 3.1.2).

### 2.1.4 In situ aerosol measurements

The Vienna air quality monitoring network, which is maintained by the "Wiener Umweltschutzabteilung (Magistratsabteilung 22)", provides continuous half-hourly values of PM2.5 and PM10 from six and thirteen, respectively, in situ instruments (e.g. Grimm EDM180) within the boundaries of Vienna (https://www.wien.gv.at/ma22-lgb/luftgi.htm). In this study, PM10 data are obtained from the station "Gerichtsgasse" (48.2611°N, 16.3969°E, 164 m a.s.l.)

located in Vienna's 21$^{st}$ district "Floridsdorf" (see Fig. 1), which is a site characteristic for the urban background and located close to the 74° azimuthal viewing direction of the BOKU MAX-DOAS. These measurements, which have been continuously performed since January 2017 using a Grimm EDM180 (Spangl, 2019), are used for the comparison of MAX-DOAS retrieved near-surface AE (see section 3.1.3).

## 2.1.5 Pyranometer

The measured global radiation, which is used for the selection of cloud-free days (see section 2.2.4), is obtained from star pyranometer (Schenk) observations, which are performed since the year 2005 at the BOKU-Met measurement platform. Briefly, the Schenk star pyranometer has six black and six white painted sectors, whereby the temperature difference between the black and white segments is proportional to the incident solar radiation.

## 2.2 Data retrieval and analysis

## 2.2.1 Vertical sensitivity, information content, and sources of errors

The vertical sensitivity of MAX-DOAS measurements, which is highest close to the surface, strongly decreases with altitude up to about 2 km, in particular for the UV channel and albeit in weaker form also for the Vis channel (e.g. Rodgers and Connor, 2003; Bösch et al., 2018; Tirpitz et al., 2021). Above that altitude, BOREAS and, in general, MAX-DOAS optimal estimation retrieval results converge with the a priori profile. Consequently, an accurate quantification of aerosols is not possible at these atmospheric layers. In contrast to ceilometer observations, the vertical resolution of MAX-DOAS is limited at the surface (~100 m) and increases with altitude, affecting both the profile shape and the near-surface concentrations of aerosols.

In order to make the two profiles comparable in a quantitative way, the ceilometer profiles are convoluted with BOREAS averaging kernels (AVKs), by applying the following formula introduced in Rodgers and Connor (2003): $x_{new} = x_{apri} + A(x_{ceilo} - x_{apri})$, where $x_{new}$ is the smoothed ceilometer AE profile, $x_{apri}$ and $x_{ceilo}$ denote the BOREAS a priori and ceilometer AE profile,

respectively, and A is the AVK matrix obtained from BOREAS calculations. AVKs characterize the sensitivity of the solution to the true state, and thus, the gain of information for each retrieved MAX-DOAS profile is represented by A. Degrees of freedom of signal (DOFs), which can be quantified by the trace of A, refers to the number of individual pieces of information that can be retrieved.

In Figures 2 and 3, examples of averaging kernels and associated DOFs for AE profiles retrieved with BOREAS in the UV and Vis channels, respectively, are shown for 10 October 2018. The left two panels represent the morning (06:50 UTC), the middle two panels show the noon (11:44 UTC), and the right two panels feature AVKs and associated vertical profiles for the afternoon (13:34 UTC). Overall, the vertical sensitivity is limited to about 2 km (UV) and 2.5 km (Vis) on this day, meaning that more information is obtained for the Vis channel due to longer effective light path lengths and thus, higher sensitivity. It should be noted that negative AVK values, particularly observed during noon and afternoon between 1 and 2.5 km altitude, indicate that additional aerosol loads can decrease the retrieved solution close to the surface.

In two recent studies (Friess et al., 2019; Tirpitz et al., 2021) it was found that the averaging kernels and associated DOFs of BOREAS aerosol retrievals show much smaller information content than AVKs from other retrievals developed by other groups. Nevertheless, good agreement of BOREAS vertical aerosol profiles with products of the other algorithms was found in those two studies. Low information content of AVKs is also found in our study. However, we underline that this might be related to additional regularisation terms and generally less straightforward interpretation of BOREAS AVKs. More details on BOREAS retrieved AVKs and DOFs can be found in Bösch et al. (2018) and Tirpitz et al. (2021).

In addition to the AVKs, the MAX-DOAS retrieved (black lines) and a priori (gray lines) as well as the ceilometer unsmoothed (orange lines) and convoluted (brown lines) AE profiles are depicted in Figs. 2 and 3. The error bars of the MAX-DOAS retrieved profile represent the total error of the aerosol retrieval. The total error ($S_{tot}$) can be represented in the following equation: $S_{tot} = S_{sm} + S_{fw} + S_{ns}$, where $S_{sm}$ denotes the smoothing error, $S_{fw}$ is the forward model error, and $S_{ns}$ describes the retrieval noise (Rodgers, 2004). As can be clearly seen in the two figures, aerosol profiles retrieved

in the Vis channel are less error-prone than the UV ones. More details on BOREAS errors can be found in Bösch et al. (2018).

With respect to the Vaisala CL51 ceilometer data, some uncertainties are linked with background and dark current effects. However, most of these noise aspects have been analyzed in great detail and are mostly overcome when using the most recent firmware (e.g. Kotthaus et al., 2016). Another feature found for aerosol detection by using ceilometer instruments is the effect of water vapor on the ceilometer emission wavelength. Wiegner and Gasteiger (2015), for example, found that the error in the backscatter retrieval can be in the order of 20% for mid-latitudes when water vapor absorption is ignored. It should be noted that in our study, no water vapor correction is applied.

AOD errors from AERONET sun photomters are in the range of 0.02 and 0.01 for the UV and Vis channels, respectively (Sayer et al., 2013).

## 2.2.2 Vertical AE profiles, AOD, and near-surface AE from MAX-DOAS measurements

The retrieval of AE profiles, AOD, and near-surface AE on cloud-free days is performed with the BOREAS algorithm (Bösch et al., 2018). The abundance of the oxygen molecule ($O_2$) only depends on pressure and temperature and decreases exponentially with altitude. The concentration of the $O_2$-$O_2$ collision complex ($O_4$) is proportional to the squared $O_2$ concentration and thus also decreases exponentially with altitude. The column amounts of the latter ($O_4$) can be retrieved from DOAS measurements in the UV and visible wavelength range because of its spectral absorption features (Wagner et al., 2004). In general, the BOREAS aerosol retrieval algorithm uses the difference between modelled and measured $O_4$ differential slant optical thicknesses around the $O_4$ absorption bands at 360 and 477 nm to retrieve AE profiles in an iterative Tikhonov regularization scheme (Rodgers, 2004). In more detail, the radiative transfer model (RTM) SCIATRAN is used for the computation of weighting functions, which are needed for the profile inversion of aerosols (Bösch et al., 2018). BOREAS and SCIATRAN (Rozanov et al., 2014) are linked in several ways: For the aerosol retrieval part, BOREAS uses an inversion function implemented in SCIATRAN. For the trace gas retrieval part, BOREAS calls SCIATRAN only for the RTM calculations, but not

for the inversion. In addition to the $O_4$ differential slant column densities (DSCDs) retrieved using the retrieval settings given in Schreier et al. (2020), atmospheric sondes profiles of pressure and temperature are used as input. In this study, aerosol profiling products are retrieved with BOREAS using measured atmospheric profiles of pressure and temperature from a co-located site, instead of using profiles from a U.S. Standard Atmosphere (Bösch et al., 2018; Gratsea et al., 2020) or averaged profiles of $O_3$ sonde measurements (Tirpitz et al., 2021). Atmospheric profiles of pressure and temperature used in this study are measured twice a day at the "Hohe Warte" site of ZAMG (see Fig. 1), e.g. at 12 UTC and 0 UTC. For the BOREAS retrieval, pressure and temperature profiles taken at 12 UTC, which are downloaded from a global data base (http://weather.uwyo.edu/upperair/sounding.html), are used as input.

The radiative transfer calculations with SCIATRAN are performed using the aerosol phase function and single scattering albedo of AERONET Version 3 (Almucantar Level 1.5 Inversion) data from the instrument located at BOKU, selecting the data closest in time to the MAX-DOAS measurement. The general configuration of BOREAS was used to retrieve AE values on a vertical grid ranging from the station altitude up to 4 km, with a 100 m grid step. The a priori profile was chosen to be exponentially decreasing (AE surface value: 0.18, scale height: 1.25 km) with the pre-scaling option introduced in Bösch et al. (2018) to cope with highly varying aerosol loads. The value for the scale height was determined from preliminary tests performed on measurements taken from the IUP Bremen MAX-DOAS instrument.

The period between AE profiles, AOD, and near-surface AE retrieved at recurring azimuth viewing directions was about 35-45 min until March 2019; and 50-75 minutes since then because of the added azimuthal viewing directions for the first configuration and full implementation of the second configuration (see section 2.1.1).

Although BOREAS $NO_2$ profiling products are briefly addressed to investigate the origin of aerosols over the urban environment of Vienna (see section 3.2), the main focus of this study is on aerosol profiles. Accordingly, details on the $NO_2$ retrieval are omitted and the reader is referred to Bösch et al. (2018) and Schreier et al. (2020).

### 2.2.3 Vertical AE profiles from ceilometer measurements

Range-corrected backscatter profiles (hereinafter referred to as backscatter profiles) from ceilometer observations can be converted into AE profiles as recently reported in the context of the validation of AE profiles retrieved from MAX-DOAS measurements (Bösch et al., 2018; Wagner et al., 2020). In this study, we follow the approach described in Wagner et al. (2020) to obtain AE profiles using the following procedure: In a first step, backscatter profiles are extracted from the original (daily) data files, which are made available by ZAMG. Second, extremely high values (> 100000), which appear at altitudes well above the mixing-height, are replaced with NaNs. Third, the ceilometer profiles with higher temporal and vertical resolution are aggregated to match both the time and altitude range of the MAX-DOAS AE profiles. On the one hand, gridding the data in time is achieved by finding the ceilometer measurements closest in time with the first MAX-DOAS measurements of individual vertical scans and then averaging the backscatter signals. In order to cover the duration of the MAX-DOAS vertical scan and adding a few additional ceilometer measurements before and after the individual scans, the averaging is realized over a range of five values before the start time of MAX-DOAS vertical scan and ending ten values after that time. The motivation of adding a few more ceilometer measurements before and after is to achieve better smoothing of the backscatter signals. In fact, the ceilometer averaging interval is about 9.5 minutes, while the duration of one MAX-DOAS vertical scan is about 4.5 minutes. On the other hand, gridding the data in space is achieved by averaging over intervals of ten backscatter signals (because of 10 m vertical resolution) to match the 100 m vertical sampling of MAX-DOAS measurements. Because of missing ceilometer measurements below 50 m above surface and in order to match the lowermost MAX-DOAS measurement (e.g. 260±50 m), the single 50 m above ground ceilometer measurement is used instead of averaging for the lowermost layer. The averaging is performed starting from the 360±50 m MAX-DOAS layer and up to the last layer (e.g. 3960±50 m). Effects of missing ceilometer data below 50 m on the comparisons presented below are expected to be neglegibly small in our study as the location of the ceilometer instrument is 69 m below the BOKU MAX-DOAS instrument (see Sect. 2.1). Once the ceilometer measurements are gridded to the time and vertical resolution of MAX-DOAS measurements, backscatter profiles are vertically integrated between the lowest (50 m above surface) and highest altitude (4 km). The vertically integrated backscatter profiles are scaled in an intermediate step by the AERONET AOD

at 910 nm (average of AOD at 870 nm and 1020 nm) in order to match the operating wavelength range of the ceilometer. The profiles are then scaled by the AOD at 360 nm (average of AOD at 340 nm and 380 nm) and 470 nm (average of AOD at 440 nm and 500 nm), which is in accordance with MAX-DOAS AE profiles retrieved in the UV (Arsenal and BOKU MAX-DOAS) and visible (BOKU MAX-DOAS only) spectral ranges (see Sect. 2.2.2), respectively. We note that, in contrast to Wagner et al. (2020), an extinction correction is not performed in our study because the effect of this correction was found to be negligibly small. Finally, converted ceilometer AE profiles are convoluted with BOREAS AVKs (see Sect. 2.2.1).

## 2.2.4 Selection of days with cloud-free conditions

The evaluation of UV-visible MAX-DOAS aerosol profiling products in this study is based on days with cloud-free conditions. To select cloud-free days in Vienna, the following procedure is applied:

First, clear sky global radiation for the period September 2017 to August 2019 is simulated using the RTM solver DISORT2 (Stamnes et al., 1988) of the radiative transfer software package libRadtran (Mayer and Kylling, 2005). Mean vertical atmospheric profiles (mid-latitude summer and winter as a function of the season) of atmospheric pressure and air, ozone, oxygen, water vapor, carbon dioxide and nitrogen dioxide densities are used as input parameters for the RTM calculations. The vertical profile of ozone is scaled according to the column ozone measurements taken from the WOUDC satellite database (woudc.org). Solar zenith angle (SZA) is taken at the time of MAX-DOAS measurement and AOD is taken from AERONET. The temporal resolution of the simulated global radiation matches the MAX-DOAS measurements.

Second, measured global radiation from the site of the BOKU-Met weather station (https://meteo.boku.ac.at/wetter/aktuell/) is temporally resampled to the MAX-DOAS time series. This data is compared with the simulated data of the first step in order to select cloud-free days automatically.

In the third step – the comparison – the following criteria are used to select cloud-free days: (i) data points are defined as "low error" if the difference between measured and simulated global radiation, relative to the simulated global radiation, is below or equal to 20%, (ii) for a day to be considered

as clear or partially clear sky at least 60% of the data points must be classified as "low error", and (iii) the daily sum of the second-order differences (second derivative) of the radiation time-series is used as a measure of sky condition variability. The daily sum of second-order differences of the measured data must be less than 2.5 times the simulated data (see Fig. 4). The empirical value of 2.5 was found to work best for the separation between clear and cloudy skies in our case. The above criteria were found by trial-and-error. Due to the annual variation of radiation there can not be a single empirically determined criteria. After applying these criteria, a total number of 119 days from September 2017 to August 2019 remains. However, the final number of cloud-free days used in this study is further reduced due to missing MAX-DOAS and/or ceilometer and/or sun photometer and/or atmospheric sounding observations. Thus, total numbers of 102 cloud-free days (40 days in fall, 14 days in winter, 22 days in spring, and 28 days in summer) for the BOKU UV, 74 cloud-free days (27 days in fall, 10 days in winter, 9 days in spring, and 28 days in summer) for the BOKU visible, and 81 cloud-free days (27 days in fall, 10 days in winter, 16 days in spring, and 28 days in summer) for the Arsenal MAX-DOAS instruments are selected for the retrieval of vertical AE profiles, AOD, and near-surface AE. It should be noted that the total number of days with cloud-free conditions is lower for the BOKU visible and Arsenal MAX-DOAS because their operation started later in time and also because of technical problems with the BOKU visible MAX-DOAS in spring 2019, which resulted in the loss of a couple of days.

## 3 Results and discussion

## 3.1 Evaluation of BOREAS aerosol profiling products

The performance of BOREAS in this study is evaluated by considering AE profiles, AOD, and near-surface AE retrievals in the UV and visible channels that fulfill the following criteria: (1) the absolute and relative difference between measured and simulated $O_4$ DSCDs at all individual elevation angles is less than $1000 \times 10^{40}$ $molec^2$ $cm^{-5}$ and less than 10%, respectively, (2) the maximum AOD is less than 1.0, and (3) no more than 50 iterations were needed in the retrieval. It should be noted that in some cases, the absolute and relative difference criteria can be reached although no convergence of BOREAS is found. In general, convergence is not reached if the a priori is not appropriate, for example, when the shape of the vertical profile and/or the assumed

AOD are wrong. Moreover, temporal changes in pressure and temperature can affect the BOREAS retrieval. The latter is related to the fact that a single daily pressure/temperature vertical profile taken from sonde measurements performed at noon is used in our study (see section 2.2.2). Bösch (2019), for example, analyzed temperatures larger than the U.S. standard temperature profile measured by sondes, which were taken at different daytimes on 15 September 2016 during the CINDI-2 campaign, and their implication for BOREAS profiling results. These increased temperatures are directly linked to smaller $O_4$ values, which lead to an increase in extinction for the aerosol profiles within the BOREAS retrieval. In this study, relative differences with values up to +12% were found for near-surface AE when using pressure/temperature from sondes instead of using profiles from a U.S. Standard Atmosphere.

### 3.1.1 Comparison of MAX-DOAS AE profiles with ceilometer AE profiles

BOREAS AE profiles retrieved from the UV and visible BOKU MAX-DOAS measurements taken at an azimuth angle of 74° (represented as a solid blue line in Fig. 1) are compared against AE profiles obtained from the ceilometer, which is about 2.25 km away and close to the selected viewing direction (see Fig. 1). The comparison is performed for all available cloud-free days falling into the period September 2017 to August 2019 (see section 2.2.4) and presented for the different seasons.

Overall, the MAX-DOAS AE profiles retrieved in the UV and visible range are consistent with the convoluted ceilometer AE profiles in terms of linear relationship, in particular during the fall, winter, and spring seasons with correlation coefficients of $R = 0.935$-$0.996$ (UV) and $R = 0.757$-$0.999$ (visible) (see Fig. 5 and 6, respectively). In the left panels of that figures, seasonal averages of all extinction points of MAX-DOAS and ceilometer extracted from a number ($N$) of profiles available from cloud-free days within selected time intervals are correlated with each other, where extinction profile points at all altitudes have equal weight. Overall, higher correlation coefficients are observed for the Vis profiles, except for profile comparisons performed for the afternoon. Increases of BOREAS AE between 2 and 3 km altitude, which could be related to clouds in the field of view of lower MAX-DOAS viewing directions, seems to cause these lower correlation coefficients. Smallest $R$ values are found for summer, with $R = 0.888$ (UV) and $R = 0.757$ (visible).

The largest correlation coefficients during summer are found in the early morning (6-8 UTC),

which could be related to the lower mixing-heights ($< 1$ km).

The number of UV retrievals differs from the number of visible retrievals, which can be explained

by the different length of time series (see section 2.2.4) and, at the same time, higher numbers of

flagged retrievals for the UV measurements (see section 3.1). The fact that the same number of

cloud-free summer days (28 days) are evaluated for the UV and visible BOREAS retrievals, but

more BOREAS retrievals are considered for the visible channel implies that more AE profile

retrievals are flagged as invalid in the UV channel. Quantitatively speaking, 66.6% (fall), 64.9%

(winter), 61.5% (spring), and 67.2% (summer) BOREAS retrievals fulfill the criteria (see Sect. 3.1)

in the UV channel, whereas 87.1% (fall), 78.7% (winter), 81.5% (spring), and 78.3% (summer)

successful BOREAS retrievals are obtained for the Vis channel.

The right panels of Figs. 5 and 6 depict the comparison between MAX-DOAS and ceilometer near-

surface AE data, which are representative for the lowest level of the troposphere (e.g. from the

instrument's altitude up to 100 m above). In this case, the lowest extinction points from all daytime

measurements of available cloud-free days are correlated with each other. Consequently, the

number of data points is the sum of $N$ of the five selected time spans given in the left panels plus

data points from time spans before and after the selected ones. The highest set of correlation

coefficients of $R > 0.85$ and linear regression slopes ($S$) of $S > 0.815$ & $S < 1.121$ are encountered

in the fall, winter, and spring seasons, for both spectral channels. This finding implies that the

BOREAS retrieval of near-surface AE delivers the best results during that time of the year. These

plots underline the difficulties that BOREAS has in retrieving AE profiles and near-surface AE

during summer, most probably due to (i) well-mixed air masses as indicated by maximum mixing-

heights, (ii) decreasing sensitivity of MAX-DOAS with increasing altitudes, (iii) significant lower

AODs than the AERONET ones, and (iv) profiles with box-like shapes, which are not well

retrieved with the exponential a priori used.

In Figures 7 and 8, absolute ($AE_{MAX-DOAS} - AE_{ceilometer}$) and relative differences (($AE_{MAX-DOAS} -$

$AE_{ceilometer}$) / $AE_{ceilometer} * 100$)) are presented for the vertical profile retrievals in the UV and Vis

channels, respectively. In the left panels of these figures, absolute (solid black lines) and relative

(color-coded lines) differences are shown for different altitude ranges and time intervals. For the

vertical AE profiles retrieved in the UV channel it is obvious that AE values at most altitude levels are lower for the MAX-DOAS than for the ceilometer. However, there are two exceptions where both the absolute and relative differences are positive, namely near-surface in winter and very slightly between 2 and 3 km altitude, in particular found in the afternoon, but for all seasons. As the positive differences of the latter are also observed for the Vis retrievals at these altitudes, with even higher amounts of more than 100%, we again speculate that clouds in the field of view of the lower MAX-DOAS viewing directions could be the reason. Another cause for these large deviations between MAX-DOAS and ceilometer could be related to increases in temperature at these altitude levels in the afternoon. The use of only a single temperature profile from noon time sonde measurements in the BOREAS retrievals could in principle cause differences. However, as these differences are also seen before noon, in particular for the Vis retrievals in summer, we rather relate these features to clouds. The lowest absolute and relative differences are found for the vertical AE profiles retrieved in the Vis channel for the fall and spring season.

The right panels of Figs. 7 and 8 show the distribution and mean of absolute differences obtained for the UV and Vis channels, respectively, for the near-surface level. Overall, the best agreement of near-surface AE is achieved for the spring and winter seasons, with slightly lower differences found for the Vis retrievals. It should be noted that with the exception of winter, mean absolute differences in near-surface AE are negative for all seasons.

## 3.1.2 Comparison of MAX-DOAS AOD with sun photometer AOD

The retrieved BOREAS AOD using UV (360 nm) and visible (477 nm) BOKU MAX-DOAS measurements taken at an azimuth angle of 74° is evaluated against the AOD (average of AERONET AOD at 340 and 380 nm for the UV as well as average of AERONET AOD at 440 and 500 nm for the visible channel) obtained from co-located sun photometer observations (see Fig. 9). As already found for the comparison of AE profiles, the best agreement between the two independent AOD measurements is found in fall ($R = 0.953$ and $R = 0.934$ for the UV and visible channel, respectively). While BOREAS generally underestimates AOD in the UV channel, AOD obtained in the visible channel is slightly overestimated during fall and spring. Lower BOREAS AODs are expected because of the limited sensitivity of MAX-DOAS profiling for higher altitudes,

e.g. above 4 km (Bösch et al., 2018; Tirpitz et al., 2021), whereas AERONET AODs might better

represent elevated aerosol in the free troposphere and stratosphere. The slopes of the linear

relationship are generally higher for the comparison of AODs retrieved in the Vis channel.

Interestingly, correlation coefficients and slopes for the UV and Vis retrievals are in best agreement

for the winter season. This could be an indication that aerosols are accumulated in the lowest layers,

where the sensitivity of MAX-DOAS is highest for both the UV and Vis channels. In contrast, the

largest discrepancy of slopes is found for spring, which could arise from elevated aerosols that are

better retrieved due to better sensitivity in the Vis channel. One explanation for the overestimations

noticed in the visible channel of the BOREAS retrievals in fall and spring (with slopes larger than

1) could be linked to spatial variations of AOD over the urban environment of Vienna, which will

be discussed later in section 3.2. As a consequence of different viewing geometries (e.g. 74°

azimuthal pointing with the MAX-DOAS vs. direct sun observations with the sun photometer), the

two measurements do not always sample the same air masses.

## 3.1.3 Comparison of MAX-DOAS near-surface AE with in situ surface PM10

In addition to the evaluation of BOREAS near-surface AE against ceilometer observations (see

Sect. 3.1.1), AE is compared against surface in situ measurements of particulate matter. Near-

surface AE can be extracted from both ceilometer AE profiles and MAX-DOAS/BOREAS

retrievals. The evaluation is performed through comparison with surface PM10 concentrations

obtained from the air quality monitoring station "Gerichtsgasse (Floridsdorf)", which is about 5.5

and 3.25 kilometers away from the BOKU MAX-DOAS and ceilometer, respectively (see Fig. 1).

It should be noted that AE and PM10 are two different physical quantities and thus, a perfect

correlation is not expected. In agreement with the previous findings, the BOREAS AE retrievals in

the UV and visible channels are qualitatively most consistent with ambient surface PM10

concentrations during the fall, winter, and spring seasons (see Fig. 10 and Fig. 11). For the near-

surface AE retrieved in the UV (Vis) channel, the strongest correlation with $R = 0.782$ ($R = 0.825$)

is found for the fall (spring) season. The slopes and intercepts of the linear regression characterizing

the BOREAS AE and the PM10 datasets are in very good agreement with those obtained from the

linear regression between the ceilometer AE and surface PM10 concentrations in that season of the

year, but also in winter and spring. This result highlights the strong performance of BOREAS, in particular for the lowest 100 m. It should be noted that BOKU and ZAMG sites are located in suburban areas, whereas the location of the in situ station "Gerichtsgasse" is characterized as urban (Spangl, 2019). This could explain the larger scatter observed during winter, which might be the result of a combination of spatial differences in emission strength, different measuring heights, and rather stable meteorological conditions, thus favoring less mixing of aerosols.

## 3.2 Spatial variability of AOD and near-surface AE

To better understand the spatial variabilities of AOD and near-surface AE and the origin of aerosols over the urban environment of Vienna, combined MAX-DOAS aerosol and $NO_2$ profiling products were used. Towards this direction, the MAX DOAS UV data for the five (BOKU) and six (Arsenal) azimuthal angles (see Fig. 1) were analyzed in more detail. Because of the rather weaker performance of BOREAS during summer, the results of the following analysis are presented for the fall, winter, and spring seasons only. In order to make the MAX-DOAS measurements of all viewing directions comparable, the retrieved AOD and near-surface AE together with the $NO_2$ retrieval products of tropospheric $NO_2$ vertical column density (VCD $NO_2$) and near-surface $NO_2$ concentrations, are interpolated to half-hour intervals. In a second step, only those time intervals with available aerosol and $NO_2$ columns at all eleven azimuth angles are considered.

Consequently, half-hour intervals with missing observations for at least one azimuth angle are discarded from the data sets. Inevitably, this filtering procedure further reduces the number of aerosol profiles obtained during cloud-free conditions ($N = 52$ in fall, $N = 24$ in winter, and $N = 30$ in spring). After removing half-hour intervals with missing observations, the remaining data points are averaged per azimuth angle and per season, excluding summer as mentioned before.

The relationship between the spatial variability of averaged AODs and averaged VCD $NO_2$ is presented in Fig. 12. The results reveal that higher AOD values (e.g. the vertically integrated AE) are detected by the Arsenal MAX-DOAS, whereas higher VCD $NO_2$ (e.g. the vertically integrated $NO_2$ concentration) is found by the BOKU MAX-DOAS. The reason for the lower VCD $NO_2$ values observed by the Arsenal MAX-DOAS could be that the instrument is installed on a tower

platform at 131 m above ground. Thus, $NO_2$ in the lowest layers close to the surface is not captured by the instrument. Higher AOD amounts at the Arsenal site could be related to industrial emission sources nearby. The highest amounts of both AOD and VCD $NO_2$, which includes observations from only a couple of cloud-free days, are detected by both instruments during fall. A closer look at the averaged BOKU MAX-DOAS retrieval results reveals that the ratio of the maximum AOD (74°) over the minimum AOD (144°) was 1.07 (fall), 1.13 (winter), and 1.08 (spring). For the VCD $NO_2$, the opposite trend is observed with the highest values towards the urban core and the lowest in the suburban areas in the north-east. Greater irregularities from this pattern, but still significant spatial differences are found for the Arsenal MAX-DOAS with the averaged AOD maximum (348°) in spring, being also ~10% higher than the minimum (324°).

Similarly, the spatial patterns of averaged near-surface AE, in relation to averaged near-surface $NO_2$ concentrations, are illustrated in Fig. 13. Overall, near-surface AE increases with increasing $NO_2$ concentrations, suggesting that both aerosols and nitrogen oxides (NOx) are released from anthropogenic emission sources. The highest values for both near-surface AE and $NO_2$ are found in winter, followed by fall and spring. Interestingly, the highest winter amounts of both near-surface AE and $NO_2$ are observed when the BOKU MAX-DOAS instrument is measuring at an azimuth angle of 88°. Along the respective light path of this viewing direction, heavy-traffic roads and a heat-generating power-station (waste incineration plant in Spittelau, 48.2344° N, 16.3594° E) can be found. In contrast, AE and $NO_2$ near-surface amounts are second-lowest at this viewing direction during fall and spring, which could be interpreted as an indication of a significant contribution of the heat-generating power-station to local air pollution in winter. As expected, higher spatial differences are found for the near-surface aerosol and $NO_2$ profiling products than for column-integrated retrieval results. While average BOKU MAX-DOAS retrieval results reveal that maximum near-surface AE (144°) is higher than minimum near-surface AE (74°) by a factor of 1.32 in spring, the largest relative difference between maximum (324°) and minimum (10°) averages of 25% is observed for the Arsenal MAX-DOAS retrievals in winter.

## 4 Summary and outlook

In this study, an evaluation of BOREAS aerosol profiling products is presented by comparing AE profiles, AOD, and near-surface AE retrieved from UV and visible MAX-DOAS measurements with data from co-located ceilometer, sun photometer, and in situ instruments. It is the first time that AE profiles are reported for different seasons and daytimes over the urban environment of Vienna.

Both the location and viewing direction of the BOKU MAX-DOAS are arranged in a way to cover as much as possible of the vertical extent of the measurements taken by the ceilometer, resulting in an overlap of an altitude up to 4 km. The rather short distance of 2.25 km between the BOKU MAX-DOAS and ceilometer further reduces effects that could arise from spatial variations. In addition to the evaluation of the vertical AE profiles, measurements of co-located sun photometer (a few meters away) and in situ instruments are obtained to assess the quality of vertically-integrated and near-surface BOREAS retrieval results.

In contrast to the recent BOREAS-based profile studies (Bösch et al., 2018; Gratsea et al., 2020; Tirpitz et al., 2021), this study takes into account measured atmospheric profiles of pressure and temperature taken at a co-located site of the Austrian official weather service (e.g. the same site where the ceilometer is operated). To systematically evaluate the retrieved BOREAS aerosol profiling products in Vienna, MAX-DOAS measurements from more than a hundred cloud-free days covering all seasons of the 2017-2019 period are considered.

The results of this study show that the retrieved BOREAS AE profiles from the BOKU MAX-DOAS measurements are consistent with AE profiles from the co-located ceilometer. The highest correlation coefficients of 0.936-0.996 (UV) and 0.918-0.999 (visible) are found for the fall, winter, and spring seasons. The largest discrepancies between the two independent measurements, also in terms of absolute and relative differences, arise during summer, most probably as a result of elevated mixing-heights leading to pronounced vertical mixing of air masses. The good performance of BOREAS is underlined by the agreement found when AOD and near-surface AE are compared with AERONET AOD and in situ PM10 measurements, again with the exception of summer. A summary of correlation coefficients obtained in this study is given in Table 2.

After resampling BOREAS aerosol profiling products, the spatial variability of vertically-integrated and near-surface aerosol amounts was investigated. While relative differences of the

mean AOD retrieved from MAX-DOAS measurements taken at different azimuth angles are on the order of 7-13%, larger relative differences of up to 32% between averaged values obtained for the different viewing directions are found for near-surface AE. The high correlation of the near-surface AE and near-surface $NO_2$ suggests that the aerosol layer close to the ground is mainly of anthropogenic origin. However, cases of high AOD are sometimes also found at low $NO_2$ VCDs, most probably as a consequence of trans-boundary pollution and/or dust events that temporally affect air masses above the urban environment of Vienna.

In conclusion, good agreement between MAX-DOAS aerosol profiling products and data from co-located instruments is found, highlighting the strong performance of BOREAS for the retrieval of tropospheric vertical aerosol profiles covering the range between the instrument's altitude up to 4 km as well as its capability to detect spatial variations of aerosol amounts over urban environments.

**Data availability.** Data can be requested from the corresponding author (stefan.schreier@boku.ac.at).

**Author contributions.** SFS, TB, and AR formulated the overarching goals of this study. TB performed calculations with the MAX-DOAS profile retrieval algorithmn BOREAS. SFS applied the method to convert ceilometer backscatter profiles into aerosol extinction profiles. PW applied a RTM to simulate global radiation and MR developed the procedure to select cloud-free days. SFS performed the analyses and prepared the manuscript. SFS and AR are responsible for the continuous operation of the BOKU MAX-DOAS instrument. SFS, AR, KL, and MV are responsible for the continuous operation of the Arsenal MAX-DOAS instrument. PW as the principal investigator of the AERONET Vienna_BOKU site provided the sun photometer measurements. CL is responsible for the continuous operation of the ZAMG ceilometer and provided both range-corrected backscatter and mixing-height data. All authors contributed to the writing of this manuscript.

**Competing interests.** Andreas Richter is a member of the editorial board of the journal.

**Acknowledgements.** This study was funded by the Austrian Science Fund (FWF): I 2296-N29, the German Science Foundation (DFG): Ri 1800/6-1, and A1 Telekom Austria. Special thanks go to Werner Sagmeister and Helmut Kropf from A1 Telekom for their organizational and technical support. Our thanks go to the University of Wyoming for making available the atmospheric sounding data. We would like to thank "Amt der Wiener Landesregierung" and "Umweltbundesamt" for making the air quality (e.g. PM10) data freely available. Many thanks go to the (extended) VINDOBONA team for helping to establish a MAX-DOAS measurement network in Vienna. Finally, we would like to thank the BOKU-Met weather station team for helping to maintain meteorological instruments and making freely available its data.

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

1    Table 1. Locations, data products, and characteristics of the instruments used in this study.

| Instrument | Location | Manufacturer | Data products | Temporal resolution | Start of measurements | Reference |
|---|---|---|---|---|---|---|
| MAX-DOAS (BOKU) | 48.2379°N 16.3317°E 267 m a.s.l. | custom-made | AE profiles AOD near-surface AE | 35-70 min [a] | May 2017 | Schreier et al. (2020) |
| MAX-DOAS (Arsenal) | 48.1818°N 16.3908°E 333 m a.s.l. | custom-made | AE profiles AOD near-surface AE | 35-70 min [a] | August 2018 | Behrens et al. (2019) |
| Ceilometer | 48.2483°N 16.3564°E 198 m a.s.l. | VAISALA | AE profiles | ~ 30 s | July 2012 | Lotteraner & Piringer (2016) |
| Sun photometer | 48.2379°N 16.3317°E 267 m a.s.l. | CIMEL | AOD | ~ 3min | May 2016 | Holben et al. (1998) |
| In situ monitor | 48.2611°N 16.3969°E 164 m a.s.l. | GRIMM | surface PM10 | 30 min | January 2017 | Spangl (2019) |
| Star pyranometer | 48.2379°N 16.3316°E 266 m a.s.l. | SCHENK | Global radiation | 10 min | March 2005 | - |

3    [a] temporal resolution refers to the time span between recurring elevation sequences at one specific azimuth angle

1    Table 2. Summary of correlations obtained in this study.

| R | vertical AE profiles | | near-surface AE | | | | AOD | |
| | MAX-DOAS vs. ceilometer | | MAX-DOAS vs. ceilometer | | MAX-DOAS vs. in situ | | BOREAS vs. AERONET | |
| | UV | visible | UV | visible | UV | visible | UV | visible |
|---|---|---|---|---|---|---|---|---|
| Fall | 0.971-0.986 | 0.968-0.999 | 0.890 | 0.913 | 0.782 | 0.814 | 0.953 | 0.939 |
| Winter | 0.936-0.954 | 0.918-0.989 | 0.935 | 0.865 | 0.765 | 0.739 | 0.903 | 0.874 |
| Spring | 0.963-0.996 | 0.989-0.997 | 0.869 | 0.925 | 0.777 | 0.825 | 0.799 | 0.924 |
| Summer | 0.888-0.992 | 0.757-0.995 | 0.488 | 0.613 | 0.286 | 0.369 | 0.370 | 0.505 |

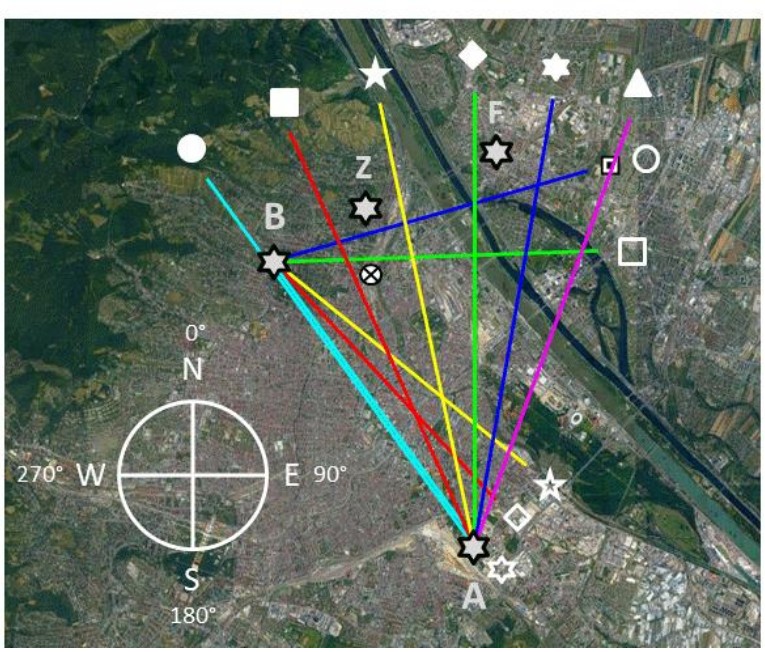

Figure 1. Geographical location of co-located instruments considered in this study: Arsenal (A)

and BOKU (B) MAX-DOAS with their associated azimuthal viewing directions (in clockwise

direction): 324° (cyan), 336° (red), 348° (yellow), 0° (green), 10° (blue), and 20° (magenta)

(Arsenal MAX-DOAS) as well as 74° (blue), 88° (green), 129° (yellow), 137° (red), and 144°

(cyan) (BOKU MAX-DOAS). The white symbols at the end of azimuthal viewing directions are

shown to better interpret the results of Figs. 12 and 13. Ceilometer (Z), sun photometer (B), and in

situ (F) instruments are located close to the 74° azimuthal viewing direction of the BOKU MAX-

DOAS. The linear distance between the two MAX-DOAS instruments is ~7.5 kilometers. The cross

in circle symbol indicates the location of the waste incineration plant. Image © Google Earth.

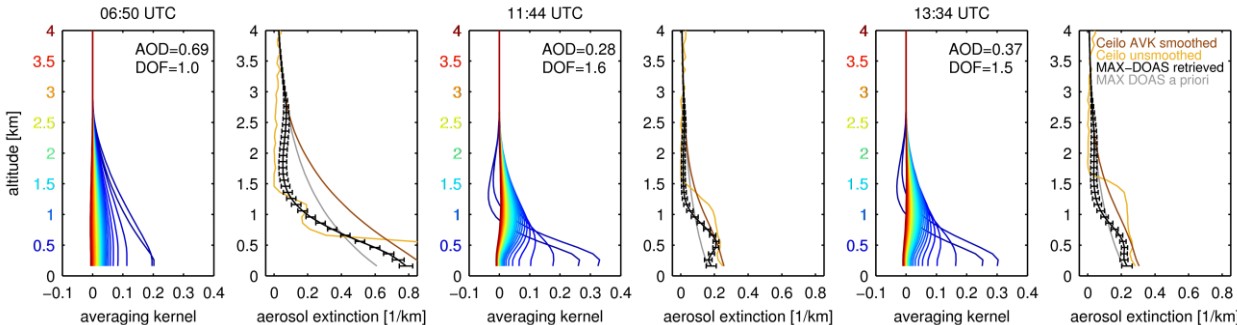

Figure 2. Exemplary averaging kernels for the retrieved aerosol extinction in the UV channel (MAX-DOAS: 338-370 nm, ceilometer: 360 nm) for the morning (06:50 UTC), noon (11:44 UTC), and afternoon (13:34 UTC) of 10 October 2018. Altitudes and corresponding AVK lines are associated with a color. Also shown are the respective MAX-DOAS a priori (gray lines), MAX-DOAS retrieved (black lines), ceilometer unsmoothed (orange lines), and ceilometer AVK smoothed (brown lines) vertical AE profiles. The error bars of the MAX-DOAS retrieved vertical AE profiles represent the total error of the BOREAS aerosol retrieval. Degrees of freedom of signal (DOFs) are given for each individual AVKs. The respective AOD values are obtained from AERONET measurements.

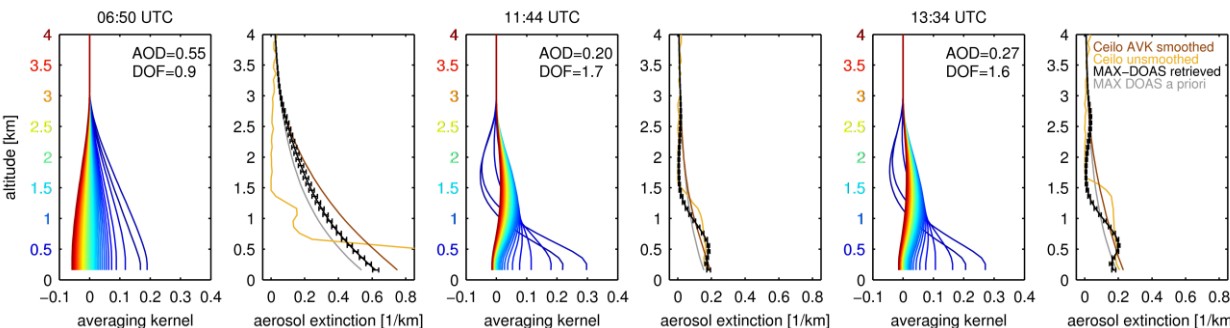

Figure 3. Same as Fig. 2, but for the visible channel (MAX-DOAS: 425-490 nm, ceilometer: 470 nm).

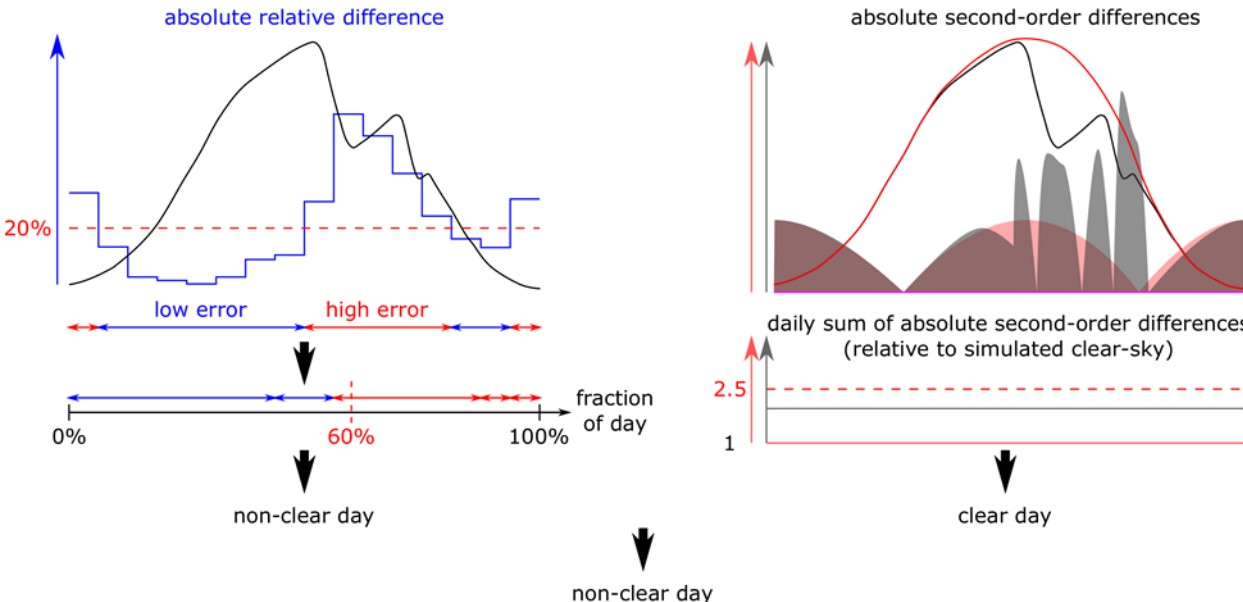

Figure 4. Criteria and process for selecting cloud-free days: (i) data points are defined as "low error" if the difference between measured and simulated global radiation, relative to the simulated global radiation, is below or equal to 20%, (ii) for a day to be considered as clear, or partially clear sky at least 60% of the data points must be classified as "low error", and (iii) the daily sum of the second-order difference is used as a measure of sky condition variability. The daily sum of the second-order difference of the measured data must be less than 2.5 times than the simulated data.

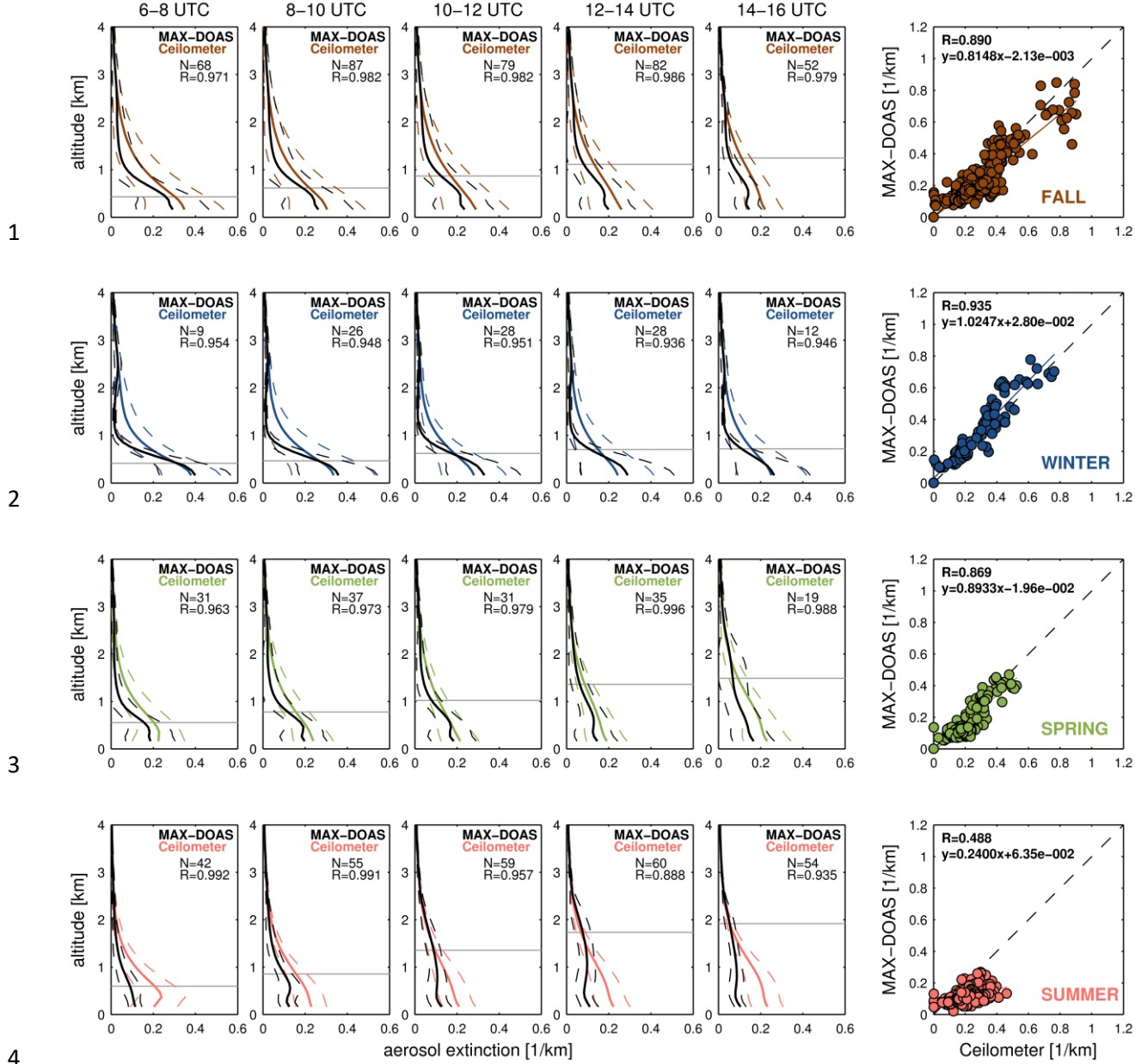

Figure 5. Averaged vertical AE [1/km] profiles retrieved from BOKU MAX-DOAS (B, see Fig. 1) at 74° azimuth angle (black solid lines) and ceilometer (Z, see Fig. 1) (color-coded solid lines) observations (left panels) for selected time periods and the four seasons fall (SON), winter (DJF), spring (MAM), and summer (JJA). The dashed black and color-coded lines represent the standard deviation of MAX-DOAS and ceilometer averaged vertical AE profiles, respectively. The gray horizontal lines illustrate corresponding averages of mixing-heights from ceilometer measurements. Scatter plots of MAX-DOAS and ceilometer near-surface AE for the different

1  seasons are shown in the right panels. Data of cloud-free days as defined in Sect. 2.2.4 between 1

2  September 2017 and 31 August 2019 are included. The measurements shown here are

3  representative for the UV channel (MAX-DOAS: 338-370 nm, ceilometer: 360 nm).

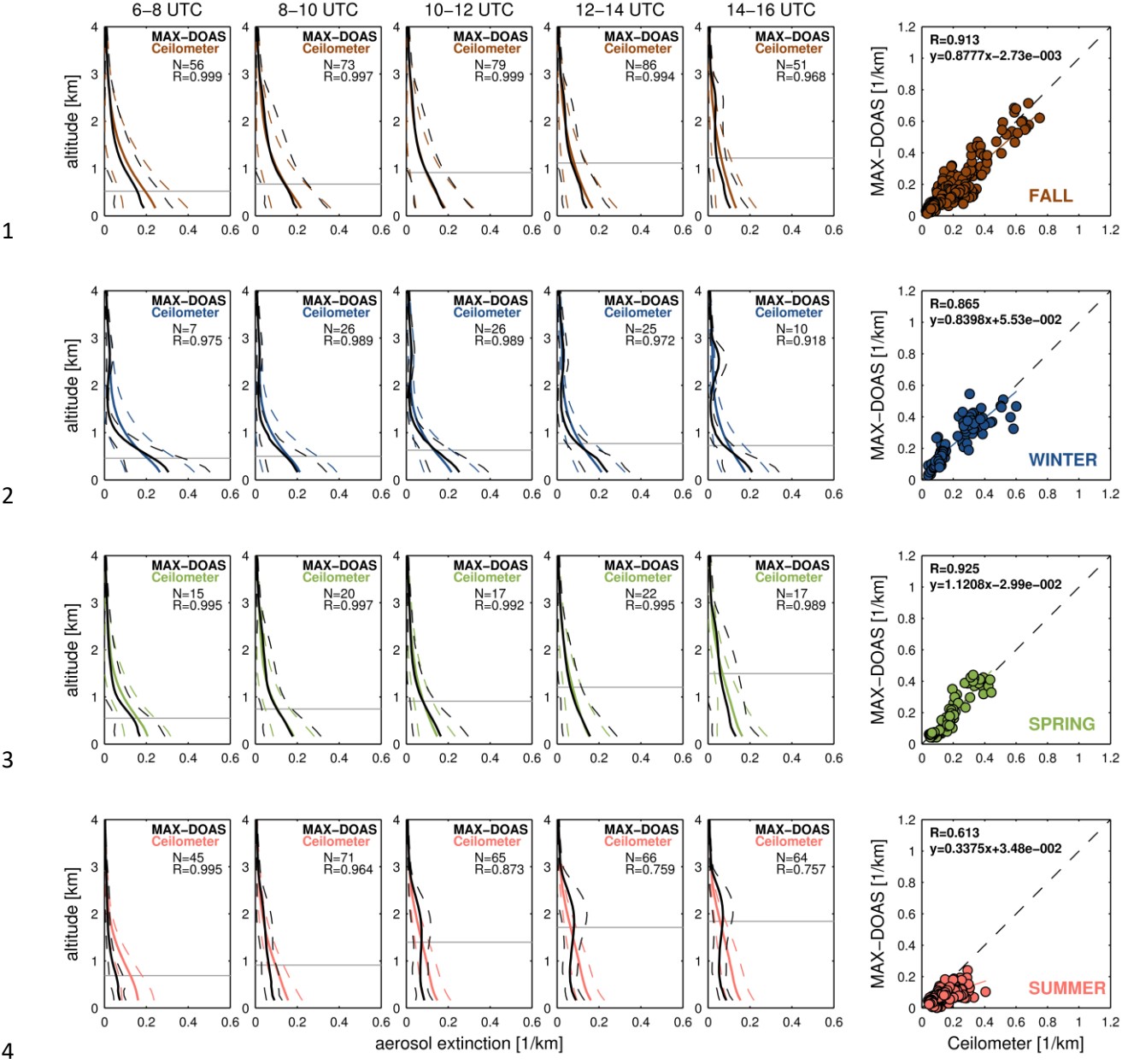

Figure 6. Same as Fig. 5, but for the visible channel (MAX-DOAS: 425-490 nm, ceilometer: 470 nm).

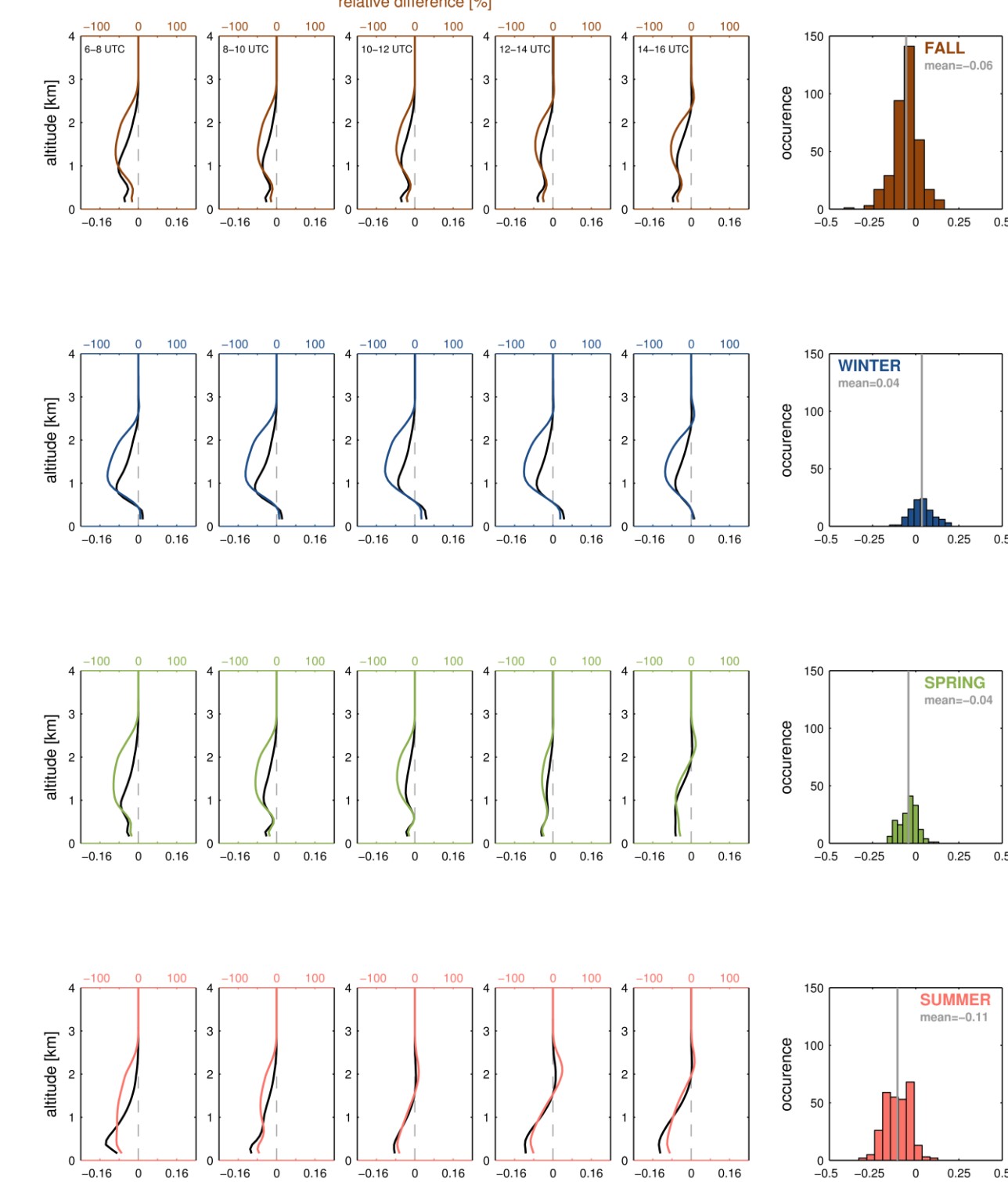

Figure 7. Absolute (black solid lines, lower axis) and relative (color-coded lines, upper axis) difference of averaged profiles obtained in the UV channel (MAX-DOAS: 338-370 nm, ceilometer: 360 nm). The distribution of absolute differences of near-surface AE for the different seasons is shown in the right panels. Data of cloud-free days as defined in Sect. 2.2.4 between 1 September 2017 and 31 August 2019 are included.

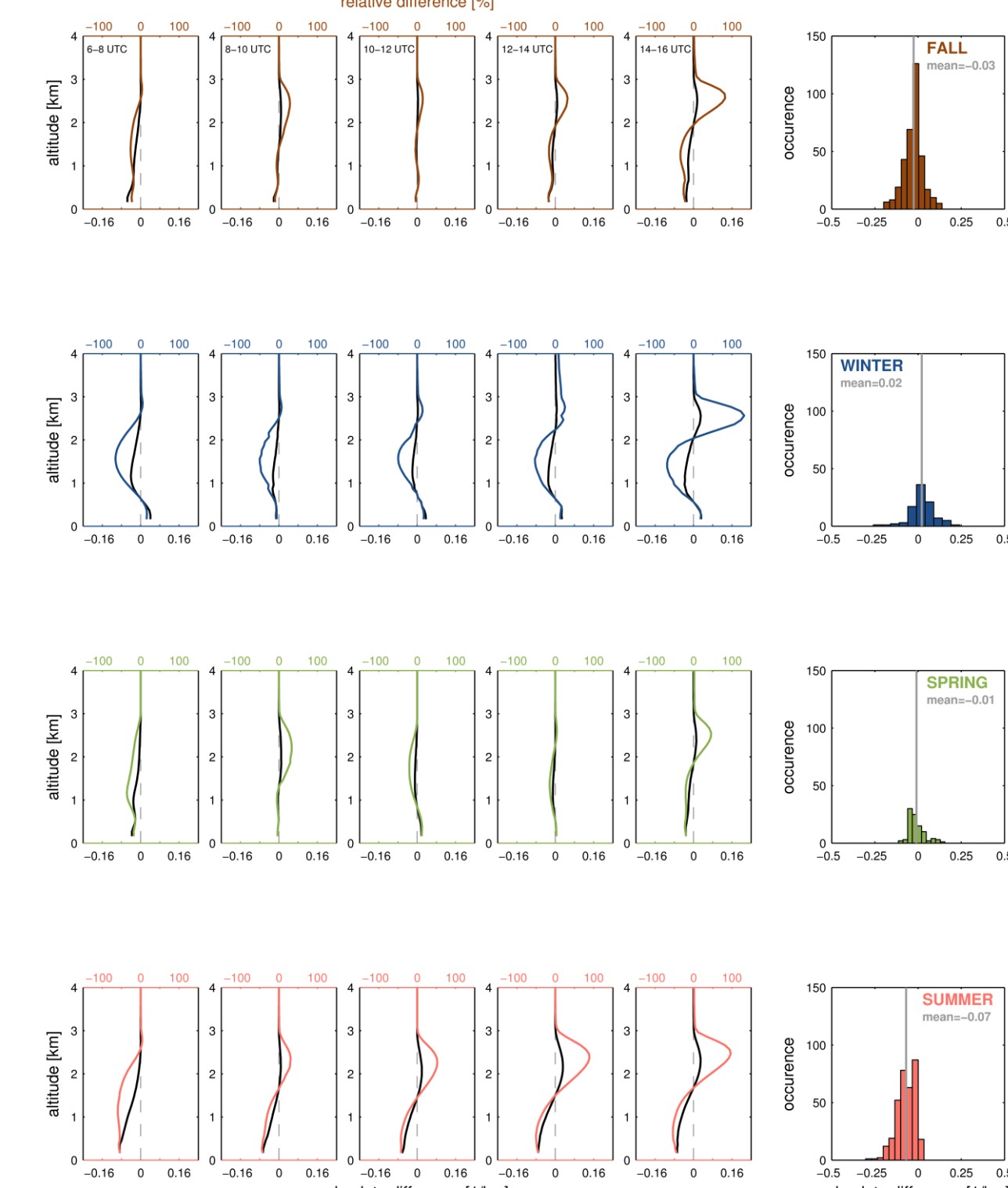

1    Figure 8. Same as Fig. 7, but for the visible channel (MAX-DOAS: 425-490 nm, ceilometer: 470

2    nm).

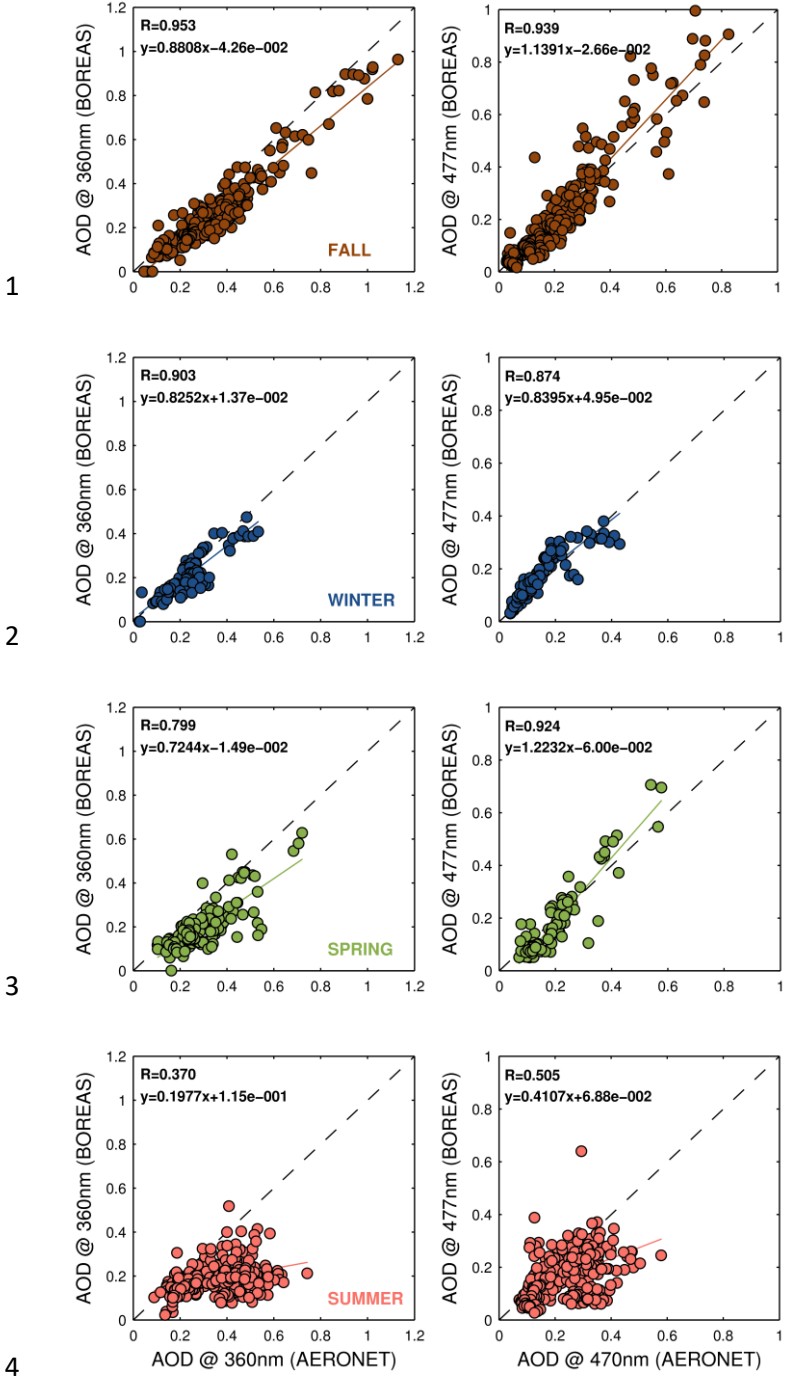

Figure 9. AOD scatterplots with their associated regression coefficients for the fall (brown), winter (blue), spring (green), and summer (red) seasons, illustrating the linear relationship of BOREAS AOD (obtained from BOKU MAX-DOAS, B, see Fig. 1) vs. AERONET AOD (obtained from sun photometer, B, see Fig. 1) in the UV (left panels) and visible (right panels) channels. Data of cloud-free days as defined in Sect. 2.2.4 between 1 September 2017 and 31 August 2019 are included.

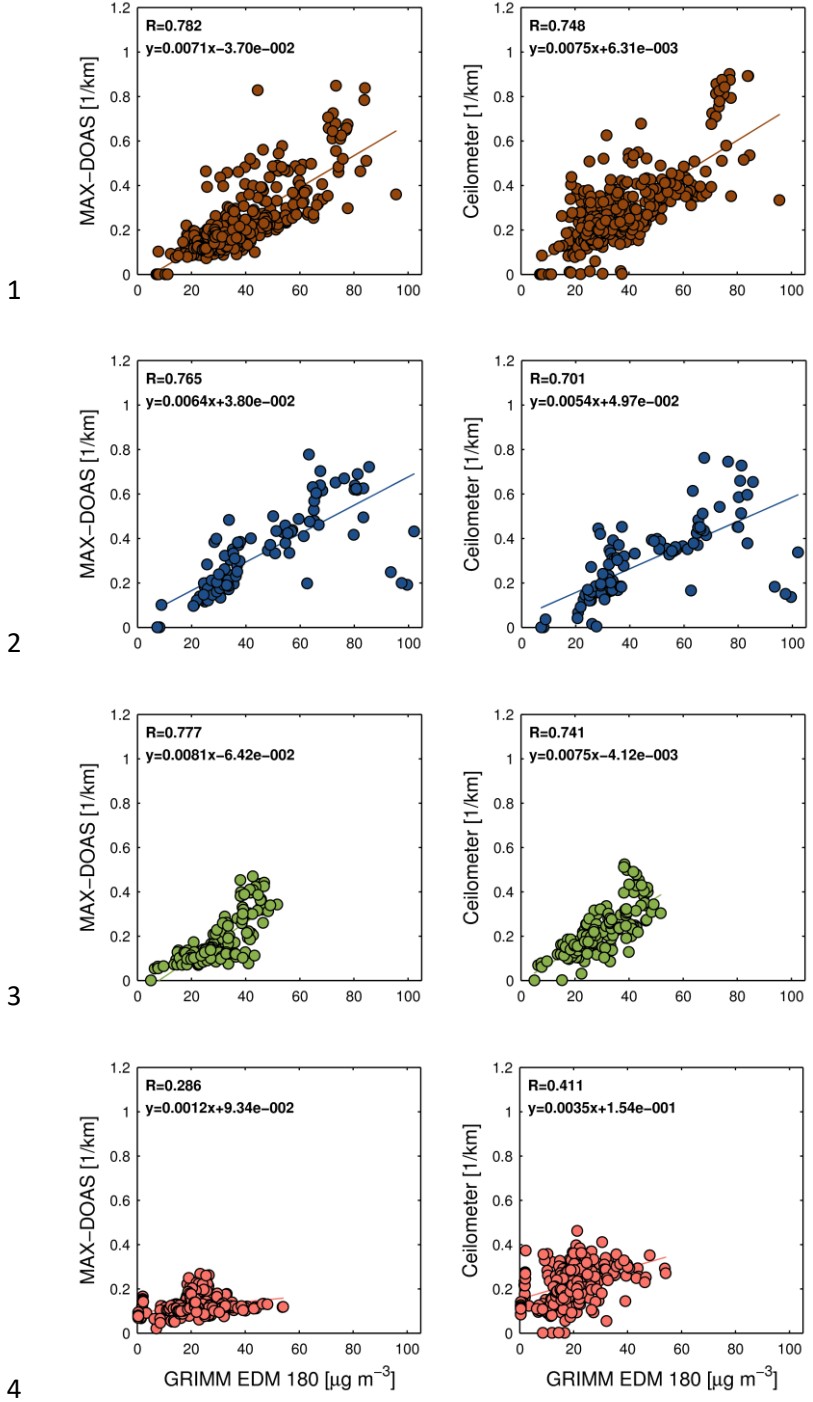

Figure 10. AE scatterplots with their associated regression coefficients for the fall (brown), winter

(blue), spring (green), and summer (red) seasons, illustrating the linear relationship of BOREAS

(B, see Fig. 1) (left panels) and ceilometer (Z, see Fig. 1) (right panels) near-surface AE [1/km]

retrieved in the UV channel vs. surface PM10 concentrations [µg m$^{-3}$] from the in situ monitoring

1  station. Data of cloud-free days as defined in Sect. 2.2.4 between 1 September 2017 and 31 August

2  2019 are included in the plots.

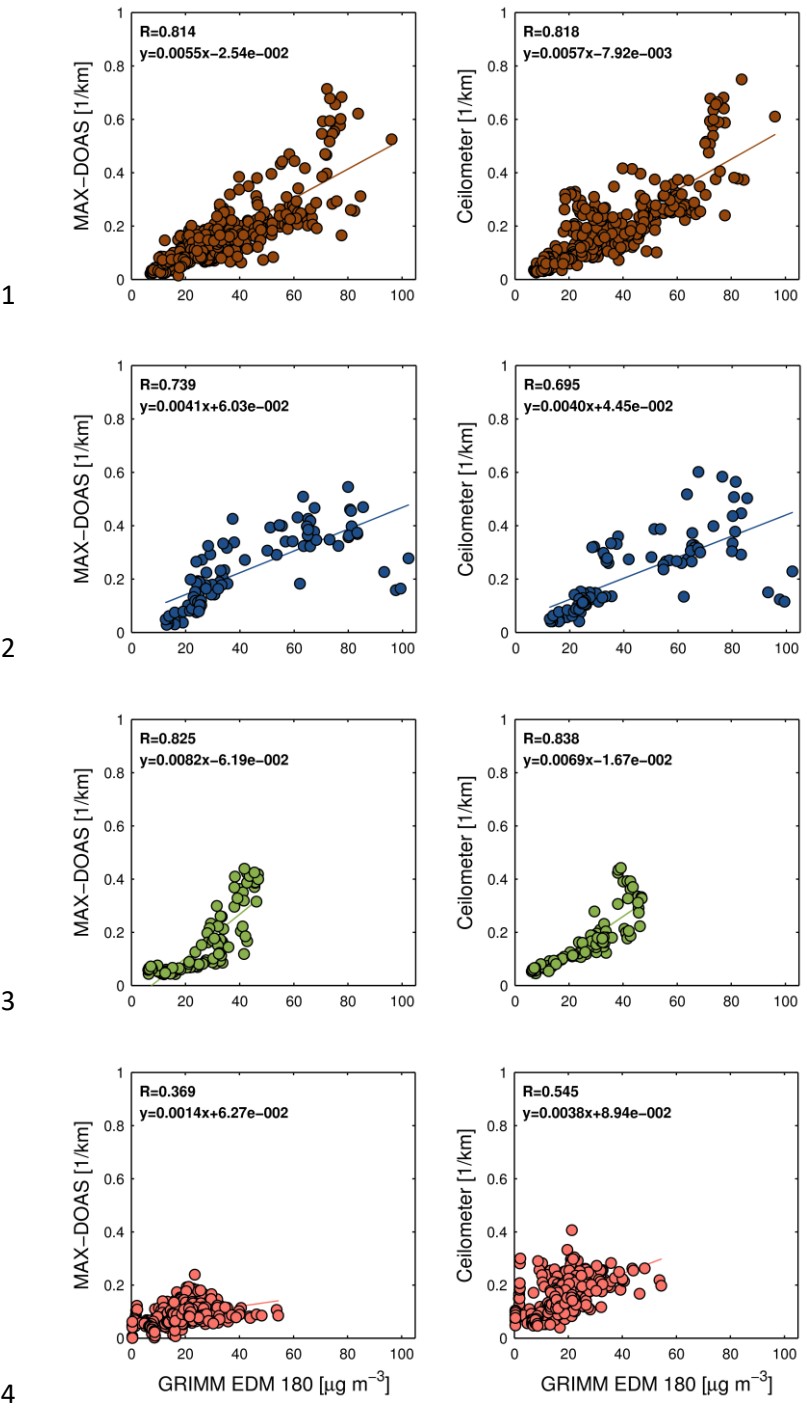

Figure 11. Same as Fig. 10, but for the visible channel.

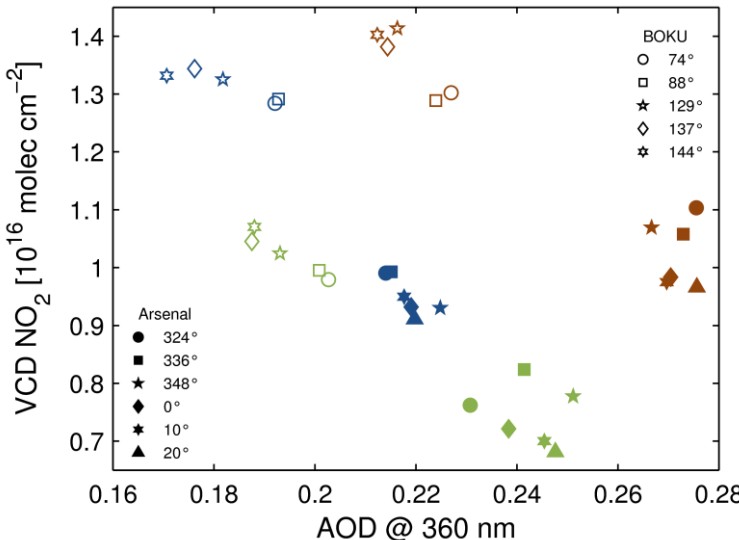

Figure 12. Spatial variability of the BOREAS vertically-integrated profiling products AOD and

VCD NO$_2$, illustrated for the seasons fall (brown), winter (blue), and spring (green) as well as for

the different azimuth angles of the two MAX-DOAS instruments. The symbols indicate azimuthal

viewing directions of the BOKU and Arsenal MAX-DOAS instruments (see also Fig. 1).

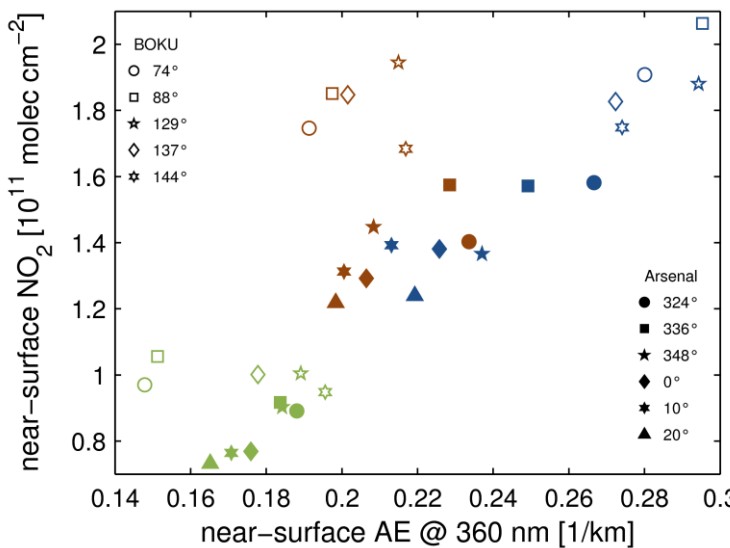

2    Figure 13. Same as Fig. 12, but for the near-surface retrieval products AE and $NO_2$.