# Peer review of "Evaluation of UV-visible MAX-DOAS aerosol profiling"

_Atmospheric Measurement Techniques, 2020_

## Author Comment (AC1)

We would like to thank the reviewer for his / her useful comments.

**GENERAL COMMENTS**

The authors have applied the BOREAS retrieval algorithm to obtain vertical profiles of aerosol extinction for a substantial data set of two different MAX-DOAS instruments in the Vienna area. They assessed the quality of these retrievals by comparing the profiles, the integrated AOD and the near-surface values with extinction profiles derived from a ceilometer, AERONET measurements and PM10 measurements respectively. The data set was large enough to be able to differentiate between the seasons. In addition they demonstrate that the two MAXDOAS instruments can be used to study the spatial variability of NO2 and aerosol over Vienna.

The paper is very well structured and clearly written. This is the first assessment of the BOREAS aerosol retrieval method with a large dataset (two instruments, and almost two years of data), and with multiple colocated comparison instruments, making it an important study.

What is missing is a discussion of the issues of this particular retrieval method found in earlier studies (Boesch et al, 2018, Tirpitz et al, 2021, Friess et al, 2019): are these issues solved, can they be confirmed or disproven?

We have now added a discussion in the new section 2.2.1 (Vertical sensitivity, information content, and sources of errors) about issues of BOREAS highlighted in recent studies (see Page 9, Line 14-21).

I am disappointed about the sole use of linear correlation coefficients to determine the quality of the retrievals, see specific comments below.

We have now introduced absolute and relative differences, in addition to the correlation coefficients (see answer to specific comments below).

**SPECIFIC COMMENTS**

The use of a linear correlation coefficient to compare vertical profiles is not obvious to me. Correlation coefficients are typically used to assess a possible linear relationship between two datasets. Do we expect a linear relationship? And if so, between what and what? It is unclear how the corellation coefficient for vertical profiles is defined in this paper. Is it defined with respect to the average profile? And are relative or absolute differences used? Do all altitudes have equal weight in this definition? Please clarify how to interpret the correlation coefficient for vertical profiles.

We agree that the sole use of a linear correlation coefficient to compare vertical profiles is not the optimal way. In our study, we have defined the correlation coefficient for vertical profiles with respect to the average MAX-DOAS/ceilometer profiles, where all altitudes have equal weight (see Page 15, Line 18-21).

In order to provide more statistics for the comparison between vertical profiles (MAX-DOAS vs. Ceilometer), we have now computed absolute and relative differences of vertical aerosol profiles and

presented these differences in additional figures (see Fig. 7 and 8) as well as in the text (see Page 16, Line 11-30 and Page 17, Line 1-2).

A more common, and more informative way to compare profiles is to look at the absolute and relative difference profiles, and quantify the differences for certain altitude ranges. Please add this to your study.

As mentioned in the answer to the comment above, we have now computed absolute and relative differences and presented these differences in additional figures (see Fig. 7 and 8) as well as in the text (see Page 16, Line 24-30 and Page 17, Line 1-15).

Please clarify the differences in sensitivity and averaging kernels between the BOREAS retrievals on one hand and the ceilometer retrievals on the other hand, and whether these differences are expected to influence the comparison.

In this study, we focus on vertical profiles within the planetary boundary layer and more generally between the station altitude and 4 km altitude. MAX-DOAS generally shows high sensitivity to lowermost layers of the troposphere. With respect to the profile retrieval algorithm BOREAS, vertical sensitivity is described by averaging kernels (AVKs). We have now introduced vertical sensitivity, information content (AVKs), and error sources in section 2.2.1.

We assume that the sensitivity of the ceilometer retrievals is more or less constant above the minimum altitude. For a better comparison of the vertical profiles from MAX-DOAS and ceilometer instruments, we have now convoluted the ceilometer data to the MAX-DOAS vertical resolution by applying the BOREAS averaging kernels (see Page 8, Line 24-27 and Page 9, Line 1-21).

It is interesting to see that the observed average profiles often show an increase towards the lowest point, both for MAXDOAS and for Ceilometer (e.g. Fig. 2, fall, 10-16UTC, spring 8-14 UTC, and even summer 8-16 UTC). Please check if there is not an intrinsic different treatment for the lowest altitude (e.g. other vertical extent, or different interpolation). If not, can you elaborate on the possible reason why this is so often seen?

It is important to note that we have used a single data point (50 m above instrument's location, which corresponds to a total altitude of 248 m asl, resulting from the instrument's altitude (198 m) + 50 m) for the lowest point of ceilometer data. This data point is assumed to be 'equal' to the lowest altitude of BOKU MAX-DOAS (260±50 m). The other data points of vertical ceilometer profiles with 10 m vertical resolution are averaged according to the 100 m vertical extent of MAX-DOAS vertical profiles (e.g. 60-140m, 150-240m, 250-340m, … etc). The different treatment of the lowest ceilometer point (using a single point instead of averaging 10 points) might explain these occasionally observed increases for the ceilometer. As we have now applied smoothing with the MAX-DOAS AVKs to the ceilometer vertical profiles, these increases disappear for the ceilometer (see Fig. 5).

Several average MAXDOAS profiles in Figure 3 (visible) show elevated aerosol between approximately 2 and 3 km altitude, while this is not seen in the ceilometer profiles. In Figure 2 (UV) this is much less pronounced. Please give an explanation, and also how this effects the observed differences.

A possible explanation for the elevated aerosol between 2 and 3 km altitude, which particularly is found for the vertical AE profiles retrieved in the Vis channel might be explained by the presence of clouds within the viewing direction of MAX-DOAS.

The following digital images of a typical elevation sequence taken on 20 August 2018 between 16:38 and 16:44 UTC indicate the presence of clouds a few kilometers away from the location of the MAX-DOAS instrument. The clouds most probably affect the retrieval of AE profiles on this day, which was defined as a 'cloud-free' day.

[Figure]

This elevated 'aerosol', which can be nicely seen in the Vis contour plot below (left panel), seems to also affect the averaged profiles presented in Fig. 3 (which after revisions is Fig. 6). While the elevated aerosol is observed for the MAX-DOAS profiles, it is not found for the ceilometer profiles, most probably because of the fact that the clouds on that specific day where not present above the instrument but rather a few kilometers away. We have now added a short discussion also in the manuscript (see Page 15, Line 21-25). In order to reduce such influence of clouds, future studies could try to find a modified filtering, e.g. by also considering the color index.

[Figure]

The regression coefficients printed in Figures 2-6 have the wrong number of significant digits. The offsets have too many, and the slopes too few. Please use the number of significant digits that is justified by the uncertainty of the coefficients. Also do not print '+-', but rather '-'.

We have now changed the number of digits as well as '+-' into '-' in the relevant figures (see Fig. 5, 6, 9, 10, and 11).

Figures 7 and 8 are very difficult to assess. I recommend adding a more map-like display. For example a figure with four subpanels for NO2, AOD, and the near-surface values, where each subpanel represents the spatial area considered and the values from fig 7 and 8 can be shown as colored bars

(or parts of a circle) in the respective viewing directions, where the length of the stripe is somewhat representative for the area observed by the MAXDOAS. The use of

We agree that Fig. 7 and Fig. 8 (which are Figs. 12 and 13 in the revised manuscript) are not very easy to assess. Using map-like figures instead is generally a good recommendation. However, as the differences of NO2, AOD and the respective near-surface values between the individual azimuthal directions are rather small, it is hard to assess these differences in a map-like figure (see unfinished example below, where the length, thickness, and color of the lines represent AOD, near-surface AE, and season, respectively).

[Figure]

We therefore would like to keep the original figures, which can be assessed in combination with Fig. 1, where we have now added the symbols to the azimuthal directions.

p2, l19: "widely documented": please give references here

We have now added a few more references (see Page 2, Line 25).

p6, l11: An elevation of 0 degree with a FOV of more than 0 will result in a vegetation signal. Is this measurement used in the retrieval?

The 0 degree elevation measurements are not used in the retrieval in our study. We have now added a sentence in the text to clarifying that measurements taken at 0° elevation angles are not used for the profile retrieval (see Page 6, Line 19-20).

section 2.2.1:

- second step to 'pre-select': what are the pre-selection criteria?

The pre-selection criteria are actually explained in the third step (see Page 13, Line 22).

- third step: this is very complex; please add a figure illustrating the procedure.

We have now added a figure illustrating the procedure (see Fig. 4).

- please explain what you mean by 'second-order difference'

This is a fixed term in mathematics: discrete equivalent of a "second derivative". https://stats.stackexchange.com/questions/351697/what-is-the-intuition-behind-second-order-differencing

We have now changed the first occurrence to: "second-order differences of the radiation time-series" (see Page 13, Line 26).

p10, l8: How large is the effect of using measured p,T-profiles instead of US standard profiles?

The effect of using p- and T-profiles instead of US standard profiles was estimated in the PhD thesis of Tim Bösch (https://media.suub.uni-bremen.de/handle/elib/1572, Page 183-186) for a single day (15 September 2016). Overall, larger (smaller) temperatures in the troposphere are directly linked to smaller (larger) $O_4$ values which will lead to an increase (decrease) in extinction for the aerosol profiles within the retrieval. More specifically, the use of US standard atmosphere led to negative relative differences, whereas the use of sonde profiles from this particular campaign day led to changes in surface extinction of up to 15% and altitude depending changes of up to 30%, in comparison to mean monthly sondes profiles used within Bösch et al., 2018. Even though the altitude depending changes due to more accurate pressure and temperature profiles might be large, the impact on mean values can be considered as small when using noon sondes profiles due to the averaging of profiles retrieved with smaller and higher temperatures, in the morning and afternoon, respectively.

Section 2.2.3: the profiles are scaled by the AOD: do you mean that the profiles are scaled so that the AOD derived from the profile matches the AERONET AOD at a specific wavelength? It is not clear to me why you need the intermediate step at 910nm

The scaling of profiles is performed by using the AERONET AOD, in accordance to the procedure described in Wagner et al., 2019, who have selected the AERONET AOD retrieved at the wavelength closest with the wavelength of the ceilometer for their scaling. This intermediate scaling in our study is performed with the average of AERONET AODs at 870 and 1020 nm, which is in good accordance with the 910 nm of the ceilometer instrument. The second scaling in our study is then performed with averages of AERONET AODs at 340 and 380 nm as well as averages of AERONET AODs at 440 and 500 nm to make the ceilometer profiles comparable to MAX-DOAS UV (360 nm) and Vis (477 nm) profiles, respectively.

Section 3.1 Only sucessful retrievals are evaluated here, which makes sense. However, it is also interesting to know how often the retrieval does not succeed (does not match the criteria). Please give numbers.

We have now calculated the percentage number of retrievals that match the criteria. The numbers are given in the text of the manuscript (see Page 16, Line 6-9).

p13, l15: I don't fully agree with this assessment. It seems that the visible retrieval tends to result in higher aerosol levels at higher altitudes, which might be the reason for the worse correlation.

After convolution of ceilometer profiles with BOREAS AVKs, correlation is better for the visible than for the UV channel. We therefore revoke the assessment (see Page 15, Line 21-22).

TECHNICAL CORRECTIONS

p2, l1: "Hile high correlation" should probably be "High correlation"?

Has been changed as suggested (see Page 2, Line 7).

p8, l11: "sza is taken from retrieved maxdoas data" is better changed to " sza is taken at the time of the maxdoas measurement"

Has been changed as suggested (see Page 13, Line 15-16).

p9, l29: is SCRIATRAN implemented in BOREAS or BOREAS in SCIATRAN?

BOREAS and SCIATRAN are linked in several ways: For the aerosol retrieval part, BOREAS uses an inversion function implemented in SCIATRAN. For the trace gas retrieval part, BOREAS calls SCIATRAN only for the RTM calculations, but not for the inversion (see Page 10, Line 24-27).

p12, l27: intervalls -> intervals

Has been changed as suggested (see Page 15, Line 20).

p14, l12-17: this sentence is too long, please split in two or three sentences.

This sentence is no longer present in the manuscript as we have deleted the discussion of 'Saharan dust' from our manuscript as this was not the main focus and rather a speculation, without enough additional data to prove.

---

## Author Comment (AC2)

We would like to thank the reviewer for his / her useful comments.

GENERAL COMMENTS

The manuscript entitled "Evaluation of UV-visible MAX-DOAS aerosol profiling products by comparison with ceilometer, sun photometer, and in situ observations in Vienna, Austria" by Schreier et al. presents vertical profiles of aerosols retrieved from Multi-Axis DOAS. The MAX- DOAS observations are compared to co-located measurements of particulate matter, AOD from sun photometer, and backscatter profiles from a ceilometer.

Aerosols play a crucial role in the atmospheric system. They affect air quality, have an impact on radiative transfer, and provide surface areas for chemical reactions. Aerosol vertical profiles from MAX-DOAS provide a valuable contribution to the understanding of the role of aerosols in the boundary layer. Therefore, this manuscript fits well into the scope of AMT.

In general, the manuscript is well written. However, some aspects of the methodology are unclear and important information is missing. In particular, the discussion mainly focuses on regression coefficients between MAX-DOAS and co-located data, but it is not always clear which quantities are actually compared in the regression analysis (see Specific Comments). While regression coefficients provide information only on the precision of the measurements, there could be more emphasis on the discussion of slope and intercept of the linear regression analysis in order to assess the overall accuracy.

We agree that the sole use of regression coefficients is not optimal for the comparison of vertical profiles. We have therefore added a discussion of slopes (see Page 17, Line 28-29 and Page 18, Line 1-5) of the linear regression analysis and moreover calculated absolute and relative differences (see answers to specific comments for more details).

Smaller correlation coefficients between surface extinction/AOD from MAX-DOAS and co-located instruments are found during summer. The authors conclude that this is due to a poorer performance of MAX-DOAS during this season. I cannot really see any reason for this. Are the DSCD errors higher or is the information content lower in summer (this would require a discussion of the averaging kernels, see my comment below)? When looking at the right panels of Fig. 2 and 3, the smaller correlation coefficients instead seem to be caused by aerosols being less abundant during summer than in other seasons, leading to poor statistics. Furthermore, there is a single outlier in the Vis data (lower-right panel of Fig. 3) that is likely to have a strong impact on the slope of the regression line.

We fully agree with the comment that aerosols are less abundant in summer than in other seasons and thus, poor statistics with respect to correlation coefficients are expected for summer. We have now introduced averaging kernels (see answer to the comment below). After convolution of ceilometer data with BOREAS AVKs, the extreme single outlier in the Vis summer data disappears and the slope is higher than before (see Fig. 6).

Apart from some general remarks on the vertical range of the retrieved profiles, a discussion on the vertical sensitivity based on averaging kernels is missing. At least some examples of averaging kernels for different atmospheric scenarios should be presented and discussed. Due to the limited vertical resolution of the MAX-DOAS extinction profiles, a quantitative comparison between MAX-DOAS and Ceilometer profiles requires the convolution of the high-resolution Ceilometer profiles with the MAX-DOAS averaging kernel according to Rodgers and Connor (2003) (see, e.g., Frieß et al., 2016 and Tirpitz et al., 2021). This is of particular importance because the vertical sensitivity of the aerosol profiles retrieved by the BOREAS algorithm appears to be limited to the lowermost 500 m (Fig. 9 in Bösch et al., 2018), which means that large fractions of the aerosol column are invisible for the MAX-DOAS instrument.

We have now introduced averaging kernels. We now present three examples of AVKs (morning, noon, and afternoon) for the UV and Vis channel for one exemplary day (10 October 2018) with changing aerosol load throughout the day (see Fig. 2 and 3 as well as Page 8, Line 16-27 and Page 9, Line 1-21). The quantitative comparison between MAX-DOAS and ceilometer vertical AE profiles is now more representative as we have now convoluted the high-resolution ceilometer profiles with the BOREAS averaging kernels according to Rodgers and Connor (2003) (see Fig. 5 and 6 as well as section 3.1.1).

Furthermore, we would like to emphasize that the sensitivity of MAX-DOAS profiling algorithm is not limited to the lowermost altitudes but it is the highest here. The sensitivity is good enough to retrieve elevated layers in synthetic data if the layer dominates the aerosol profile (see Bösch, 2019).

SPECIFIC COMMENTS

Abstract L20-23: It is not correct that coincident measurements of temperature and pressure profiles were used here for the first time for profile inversion, see e.g. Friedrich et al. (2019), who used daily radiosondes.

We have now deleted "For the first time," in the abstract (see Page 1, Line 20-22).

P6, L15: Do you use the 0° elevation measurements for the profile retrieval? I could imagine that this leads to difficulties in the inversion, either due to blocking by surrounding buildings or trees, or due to the fact that the field of view includes both atmosphere and surface.

The 0 degree elevation measurements are not used in the retrieval in our study. We have now added a sentence in the text to clarifying that measurements taken at 0° elevation angles are not used for the profile retrieval (see Page 6, Line 19-20).

P8, L12: Mean vertical profiles of which quantities are used as input parameters for the RTM?

We have now added a sentence to clarify, which quantities are used (see Page 13, Line 11-15).

I suggest to structure the sub-sections of Section 2.2 in the order of their importance, starting with the MAX-DOAS profile retrieval, and to move Section 2.2.1 on cloud flagging to the end of section 2.2.

We have now re-structured the sub-sections of Section 2.2 accordingly.

P8, L16: It is not clear to me what the term 'Hence' refers to. The term 'Differential slant column' is not defined yet and should either be explained here, or replaced by 'measurements'.

We have now deleted the term 'Hence' and replaced 'DSCDs' with 'measurements' (see Page 13, Line 16-17). The term DSCD is explained for the first time at Page 10, Line 27-29.

P8, L19-24: The technical description of the pyranometer should be moved to section 2.1.

We have now moved the technical description of the pyranometer to section 2.1 (see Page 8, Line 7-12).

P9, L4: What do you mean with 'daily total second-order difference'? Is this the mean of the second-order differences? Please clarify.

We have now introduced the term 'daily sum of the second-order differences', this should be more clear (see Page 13, Line 25-28).

P9, L25: The term 'oxygen dimer' should be avoided; O4 represents the O2 collision complex.

We have now avoided the term 'oxygen dimer' (see Page 10, Line 15-17).

P10, L7: Again, this is not the first time that measured atmospheric profiles of pressure and temperature from a co-located site are used for MAX-DOAS profile retrieval, see general comments.

We have now deleted "for the first time" (see Page 10, Line 28-29).

Section 2.2.2: An error discussion regarding the retrieved aerosol profiles is completely missing. Retrieval errors and vertical resolution based on averaging kernels, as well as information content should be discussed here. A discussion of averaging kernels is particular importance because the lack of sensitivity for aerosols at high altitude might explain parts of the discrepancies between MAX-DOAS and ceilometer, and because the ceilometer profiles should be convoluted with the MAX-DOAS averaging kernels in order to perform a quantitative intercomparison between both data sets (see general comments).

We have now introduced and discussed vertical sensitivity, averaging kernels as well as retrieval errors in our manuscript (see section 2.2.1 as well as Fig. 2 and 3). The ceilometer profiles have now been convoluted with the MAX-DOAS averaging kernels to perform a more quantitative comparison between MAX-DOAS and ceilometer (see section 3.1.1 as well as Fig. 5 and 6).

P11, L7: In which way has time been extracted from the backscatter profiles? Do you mean the time stamp of the profiles? This would not really be worth mentioning.

Yes, with time we mean the time stamp of the profiles. We have now deleted the term 'time' (see Page 12, Line 3-4).

P11, L13, 19 and 22: I think the term 'assimilation' is inappropriate here because it has a well-defined meaning in atmospheric science, namely to adapt a modelled atmospheric state to observational data in a statistically optimal way. Maybe gridding the data (in time and space) would be a more appropriate term.

We have changed the text as suggested (see Page 12, Line 7-8 and Page 12, Line 16).

P11, L15-19: Please add a sentence motivating why the temporal averaging has been done in such a quite complicated way, instead of just averaging over the duration of a MAX-DOAS scan.

The motivation behind our temporal averaging is the fact that we wanted to achieve a slightly higher number of ceilometer data (a couple of measurements before and after the MAX-DOAS elevation scan) for averaging than only the duration of the MAX-DOAS vertical scan. We have now changed the respective sentence in the manuscript (see Page 12, Line 12-15).

Please describe how you are dealing with the missing aerosol information from the Ceilometer in the lowermost 50 m, where MAX-DOAS is most sensitive and variability in aerosol extinction is probably highest. What kind of extrapolation did you apply for the calculation of the extinction in the lowermost retrieval layer and for the determination of the AOD?

It should be noted here that the location of the ceilometer (198 m a.s.l.) is about 70 meters lower than the one of the BOKU MAX-DOAS (267 m a.s.l.) because of local topography. This information is also given in the manuscript (see Sect. 2.1). In order to make the lowermost 100 meters of MAX-DOAS (station level, e.g. 260±50 m) comparable with the lowermost available ceilometer measurement, we have used the lowermost measurement of the ceilometer as a single value. We therefore did not perform any extrapolation for the layers below 50 m above the ceilometer and just assume that the single value of 50 meters above ground represents the average of the lowermost 100 meters value of MAX-DOAS, which of course is a simplification. We have added this information in the text (see Page 12, Line 18-24). AOD is only determined from MAX-DOAS measurements in our study and not from ceilometer measurements. For the scaling of ceilometer measurements, AOD from AERONET is used (see Page 12, Line 26-31 and Page 13, Line 1).

P12, L25: It is not clear what kind of data has been used for the calculation of the correlation coefficients. Did you correlate extinction at all heights, or just at the surface? Seasonally averaged or individual profiles?

In our study, we have defined the correlation coefficient for vertical profiles with respect to the seasonally averaged MAX-DOAS/ceilometer profiles for the different time intervals (e.g. 6-8 UTC, 8-10 UTC, …), where all altitudes have equal weight (see Page 15, Line 18-21).

P14, L11: Here it is stated that there is limited sensitivity of MAX-DOAS aerosol profiles above 4 km. When looking at Fig. 9 in Bösch et al. (2018), the sensitivity rather seems to be restricted to the lowermost 500 m only. Again, this means that a convolution of the Ceilometer profiles prior to the comparison is crucial – see general comments.

The sensitivity is not 'restricted' to the lowermost 500 m only. It rather is stronger below 500 m and lower above 500 m. A convolution of ceilometer profiles with the averaging kernels has now been performed (see answer to the comment above).

 Here you distinguish between the availability of total columns and surface values. Shouldn't the total column always be available if the surface value is available and vice versa, since both are derived from the according vertical profile?

Yes, if the surface value is available then also the total column value is available and vice versa. We have now changed in the text accordingly (see Page 19, Line 15).

TECHNICAL CORRECTIONS

Title of 2.1.4: 'In situ' -> 'In situ aerosol measurements'

Has been changed as suggested (see Page 7, Line 23).

P9, L23: Remove 'Briefly' from the beginning of the sentence.

We have removed 'Briefly' (see Page 10, Line 14).

P9, L26: Remove comma after 'thus'.

We have removed the comma after 'thus' (see Page 10, Line 16).

P12, L8: A threshold in difference between modelled and measured O4 DSCD of 1000 molec2/cm5 is extremely small, given that typical O4 DSCDs are in the order of 1043 molec2/cm5. Is this a typo?

Yes, this was a typo. We have changed the text as suggested (see Page 14, Line 21).

REFERENCES

Rodgers, C. D. and Connor, B. J.: Intercomparison of remote sounding instruments, J. Geophys. Res, 108(D3), 4116–4229, doi:10.1029/2002JD002299, 2003.

We have added the reference (see Page 27, Line 7-8).

---

## Author Comment (AC3)

We would like to thank the reviewer for his / her useful comments.

GENERAL COMMENTS

In this paper, aerosol extinction (AE) profiles, aerosol optical depth (AOD), and near-surface AE are retrieved from MAX-DOAS measurements acquired on cloud-free days during the September 201-August 2019 period at two stations in the vicinity of the Vienna (Austria) city centre. These retrievals are performed using the Bremen Optimal estimation REtrieval for Aerosols and trace gaseS (BOREAS) algorithm and are evaluated against co-located ceilometer, sun photometer, and in situ instrument observations covering all four seasons. The retrieved AE profiles are found to agree well with those from the co-located ceilometer in fall, winter, and spring, with correlation coefficients ranging between 0.85 and 0.99. During those seasons, a good agreement is also obtained with the ceilometer for the near-surface AE and between the MAX-DOAS and sun photometer AODs. The MAX-DOAS retrieval results appear to be less reliable in summer and the possible origins of the lower performance of BOREAS in those conditions are discussed. Finally, the spatial variability of AOD and near-surface AE over Vienna is assessed by analyzing the retrieved BOREAS aerosol profiling products in the different azimuthal pointing directions of the two MAX-DOAS instruments and for the different seasons.

This paper is well written and clearly structured and presents interesting results which fit well with the scope of AMT. I recommend the final publication of the manuscript after addressing the following major and specific comments:

Major comment: To my opinion, the present study suffers from two weaknesses: no uncertainty budget is presented for any of the MAX-DOAS retrieved quantities (AE profile, near-surface AE, and AOD) and there is no estimation and characterisation of the vertical sensitivity of the MAX-DOAS AE profile retrievals through the calculation and examination of the averaging kernels and corresponding DOFS. The uncertainties on the Ceilometer AE profiles and other ancillary data deserve also to be discussed. Both aspects (uncertainty and vertical sensitivity) of this major comment should be addressed in the revised manuscript.

In the revised manuscript, we have introduced BOREAS averaging kernels, presented and discussed examples of those and furthermore used AVKs for smoothing the ceilometer AE profiles to make them more comparable in a quantitative way (see Sect. 2.2.1). In addition to the characterization of MAX-DOAS vertical sensitivity through AVKs, we now report on errors of BOREAS retrievals, ceilometer, and sun photometer measurements (see Sect. 2.2.1).

SPECIFIC COMMENTS:

Page 3, lines 11-13: I think here it is worth explicitly mentioning that a large part of the sun photometer measurements effort is endorsed by the AERONET network. The AERONET http link could be also added. Then, no need to add the http link again on page 7, lines 7-8.

We fully agree with this comment and have changed the text accordingly (see Page 3, Line 13-15).

Page 10, line 20: the choice of the a priori scaling height (1.25km) should be justified here. Is it based on a sensitivity study using different a priori scaling height values?

This value was determined from preliminary tests performed on measurements taken from the IUP Bremen MAX-DOAS instrument. We have now added a sentence with this piece of information (see Page 11, Line 14-16).

Page 12, lines 7-9: The level of agreement between measured and simulated O4 DSCDs is used to select valid MAX-DOAS AE profile retrievals. I guess that the criteria used (absolute and relative difference smaller than 1000 molec2 cm-5 and less than 10%, respectively) is applied individually to all the elevation angles. I think this should be explicitly mentioned in the text.

Yes, the criteria used is individually applied to all elevation angles. We have now added a corresponding sentence in the text (see Page 14, Line 19-21).

Page 12, lines 13-14: It is said that temporal changes in pressure and temperature can affect the BOREAS retrieval. This general statement requires some explanation: how large can be this effect on a daily basis since daily atmospheric temperature and pressure profiles are used as input for the AE profile retrievals?

In the PhD thesis of Tim Bösch it was estimated that on a daily basis, the near-surface AE can be up to 12% larger when pressure/temperature is taken from sondes instead of using pressure/temperature profiles from the US standard atmosphere (Bösch, 2019, https://media.suub.uni-bremen.de/handle/elib/1572?locale=de). We have now added this information in the manuscript (see Page 14, Line 27-29 and Page 15, Line 1-6).

Page 13, lines 13-16: the higher correlation coefficient values between MAX-DOAS UV and ceilometer AE profiles is explained by the fact that in the UV, the MAX-DOAS instrument probes air masses closer to the ceilometer which is located at 2.25km from the MAX-DOAS. Is it a valid argument since the effective horizontal distance representative of the MAX-DOAS measurements of O4 in the UV can be as large as 10-15km or even more under clear-sky conditions? Was there any attempt to estimate the effective horizontal distances for both the UV and VIS channels?

The effective horizontal distances for the UV and Vis channels have been estimated in a recent study (Schreier et al., 2020, https://doi.org/10.1016/j.aeaoa.2019.100059) for the Vienna MAX-DOAS instruments. For measurements taken at 1° (UV) and 3° (Vis) elevation angles, mean effective horizontal distances were estimated at about 6-12 km and about 7-18 km for the UV and Vis channels, respectively.

As we have now convoluted ceilometer AE profiles with BOREAS AVKs, the statement that UV MAX-DOAS profiles are better correlated with ceilometer profiles than the Vis ones is no longer correct. It is now rather the other way round. As both UV and Vis effective horizontal distances are larger than 2.5 km, we reject our argument. We have now changed the statement in the text accordingly (see Page 15, Line 21-25).

Page 13, lines 25-29: According to the authors, BOREAS has difficulty in retrieving AE profiles and near-surface AE during summer. One possible reason is that during that season, AE profiles have a box-like shape and are therefore not well retrieved with the exponential a priori used. Did the authors make a sensitivity test using a priori profiles with box-like shape in order to see whether it improves the AE profile retrievals in summer? If it significantly improves their retrievals but also the agreement with the ceilometer and the sun photometer, I think the authors should consider to include such a sensitivity test in their manuscript.

Box-like profile shapes are problematic for MAX-DOAS inversion algorithms relying on Bayesian approaches like optimal estimation or iterative approaches as Levenberg-Marquardt (LM) and Newton-Gauß (NG) with Tikhonov terms for several reasons. 1. For most algorithms, the a priori covariance matrix is calculated as deviation from an a priori state. This means that the covariance matrix has "Zero-elements" in altitudes where the box profile is zero, which will lead to an instability during inversion. 2. Even if these covariance values are set to finite values, large changes from this box-shape are not possible within one inversion (OEM) or small numbers of iterations for LM and NG as changes are done relative to the a priori state. 3. This means that especially the box width/height needs to be known well because large deviations from this shape are not possible.

However, box-like profiles can in principle be retrieved with exponential a priori profiles resulting in oscillating features around the true box shape, with smaller AOD in layers with true finite aerosol concentration due to the a priori based additional aerosol abundance for altitudes above the true box.

Page 14, lines 9-17: According to the authors, lower BOREAS AODs are expected because of the limited sensitivity of MAX-DOAS profiling for higher altitudes, while AERONET AODs better represent elevated aerosol in the free troposphere (and stratosphere). The authors then argue that in Spring, Saharan dust events over Austria could potentially explained higher AERONET AOD, mentioning the detection of such events in the Austrian Central Alps. I think this explanation is very speculative and requires further investigation: for instance, have such events been detected during the period where the MAX-DOAS measurements presented in this study were performed? If yes, is it statistically significant, i.e. during how many days Saharan dusts stayed above the Vienna area?

We agree that this explanation is speculative. As we are focusing on vertical AE profiles below 4 km in this study, and because Saharan dust events are not the focus of this study, we decided to remove the discussion of 'Saharan dust' events from our study.

We have now added a discussion of slopes of the linear relationship between UV/Vis BOREAS retrieved and AERONET AODs (see Page 17, Line 28-29 and Page 18, Line 1-5).

TECHNICAL CORRECTIONS:

Page 12, line 27: 'intervalls' -> 'intervals'

Has been changed as suggested (see Page 15, Line 20).

Page 14, line 12: 'tropsphere' -> 'troposphere'

Has been changed as suggested (see Page 17, Line 28).